# Negative Label Guided OOD Detection with Pretrained Vision-Language Models

**Xue Jiang**[1,2]   **Feng Liu**[3]   **Zhen Fang**[4]   **Hong Chen**[5]   **Tongliang Liu**[6,7]
**Feng Zheng**[1*]   **Bo Han**[2]
[1]Southern University of Science and Technology   [2]TMLR Group, Hong Kong Baptist University
[3]TMLR Group, University of Melbourne   [4]University of Technology Sydney
[5]Huazhong Agricultural University   [6]Mohamed bin Zayed University of Artificial Intelligence
[7]Sydney AI Centre, University of Sydney
{csxjiang, bhanml}@comp.hkbu.edu.hk, feng.liu1@unimelb.edu.au,
zhen.fang@uts.edu.au, chenh@mail.hzau.edu.cn,
tongliang.liu@sydney.edu.au, f.zheng@ieee.org

## Abstract

*Out-of-distribution* (OOD) detection aims at identifying samples from unknown classes, playing a crucial role in trustworthy models against errors on unexpected inputs. Extensive research has been dedicated to exploring OOD detection in the vision modality. *Vision-language models* (VLMs) can leverage both textual and visual information for various multi-modal applications, whereas few OOD detection methods take into account information from the text modality. In this paper, we propose a novel post hoc OOD detection method, called NegLabel, which takes a vast number of negative labels from extensive corpus databases. We design a novel scheme for the OOD score collaborated with negative labels. Theoretical analysis helps to understand the mechanism of negative labels. Extensive experiments demonstrate that our method NegLabel achieves state-of-the-art performance on various OOD detection benchmarks and generalizes well on multiple VLM architectures. Furthermore, our method NegLabel exhibits remarkable robustness against diverse domain shifts. The codes are available at https://github.com/tmlr-group/NegLabel.

## 1 Introduction

In open-world scenarios, deploying machine learning models faces a critical challenge: how to handle data from unknown classes, commonly referred to as *out-of-distribution* (OOD) data (Hendrycks & Gimpel, 2017). The presence of OOD data can lead to models exhibiting overconfidence, potentially resulting in severe errors or security risks. This issue is particularly pronounced in critical applications, such as autonomous vehicles and medical diagnosis. Therefore, detecting and rejecting OOD data plays a crucial role in ensuring the reliability and safety of the model.

Traditional visual OOD detection methods (Hsu et al., 2020a; Wang et al., 2021b; Huang et al., 2021; Sun et al., 2021; Wang et al., 2021a) typically rely solely on image information, ignoring the rich textual information carried by labels. *Vision-language models* (VLMs) can leverage multimodal information, which is also beneficial for OOD detection. Some recently proposed methods attempt to design dedicated OOD detectors for VLMs. Specifically, ZOC (Esmaeilpour et al., 2022) defines the new task – zero-shot OOD detection, and uses a trainable captioner to generate candidate OOD labels to match OOD images. However, when dealing with large-scale datasets encompassing a multitude of *in-distribution* (ID) classes, like ImageNet-1k, the captioner may not generate effective candidate OOD labels, resulting in poor performance. MCM (Ming et al., 2022a) uses the maximum logit of scaled softmax to identify OOD images. However, MCM only employs information from the ID label space and does not effectively exploit the text interpretation capabilities of VLMs. Therefore, there exists untapped potential for enhancing OOD detection utilizing VLMs.

In this paper, we design a method to leverage better the knowledge carried by VLMs for OOD detection. To achieve this, we introduce a large number of negative labels that enable the model to

---

*Correspondence to Feng Zheng (f.zheng@ieee.org)

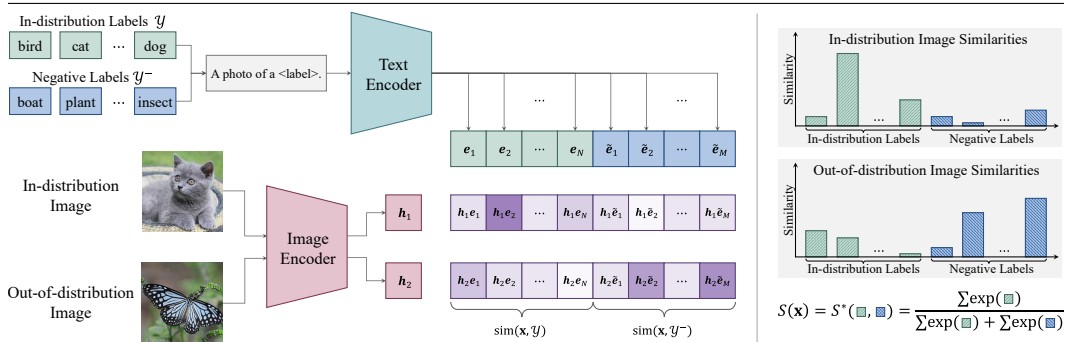

Figure 1: Overview of NegLabel. The image encoder extracts input images into image embeddings $\boldsymbol{h}$. The text encoder extracts ID and negative labels into text embeddings $\boldsymbol{e}$. All encoders are frozen in the inference time. The negative labels are selected by NegMining (see Section 3.1) from a large-scale corpus. The image-text similarities are quantified through $\boldsymbol{h} \cdot \boldsymbol{e}$, represented by purple blocks (darker shades indicating higher similarity). The right part illustrates that ID images tend to produce lower similarities with neagtive labels than OOD images. Our NegLabel score (see Section 3.2) fuses the similarities of image-ID labels (green) and image-negative labels (blue).

distinguish OOD samples in a more nuanced and detailed manner. Our proposed method NegLabel detects OOD samples by examining their affinities between ID labels and negative labels. As shown in Fig. 1, ID samples have higher OOD scores than OOD samples due to the affinity difference.

Specifically, we design the NegMining algorithm to select high-quality negative labels from extensive corpus databases. This algorithm utilizes the distance between a negative label and ID labels as a metric to assess their adequacy in terms of semantic divergence. By selecting negative labels with sufficient semantic differences from ID labels, the algorithm enhances the separability between ID and OOD samples. Additionally, we introduce a novel scheme for the OOD score, which is positively correlated with the affinities between images and the ID labels and negatively correlated with those of the negative labels. The score combines the knowledge from the ID and negative label space, thus better leveraging the VLMs' capabilities of comprehending text. Furthermore, we provide a theoretical analysis to aid in understanding the mechanism of negative labels.

Extensive experiments demonstrate that our method NegLabel achieves state-of-the-art performance on various zero-shot OOD detection benchmarks and multiple VLM architectures, like CLIP (Radford et al., 2021), ALIGN (Jia et al., 2021), GroupViT (Xu et al., 2022), and etc. Combining with CLIP, NegLabel achieves 94.21% AUROC and 25.40% FPR95 in the large-scale ImageNet-1k OOD detection benchmark (Huang et al., 2021), even surpasses numerous existing methods in fully-supervised settings. Furthermore, NegLabel exhibits remarkable robustness against diverse domain shifts. In summary, our key contributions are as follows:

- Our proposed framework, NegLabel, introduces massive negative labels to boost OOD detection. The extended label space provides a novel perspective to distinguish OOD samples, leveraging extra clues by examining the similarities between images and the extended labels.

- We propose the NegMining algorithm (see Section 3.1) to select high-quality negative labels to enhance the distinction between ID and OOD samples. Based on the picked negative labels, we design a new scheme for the post hoc OOD score (see Section 3.2) that effectively leverages the knowledge encoded in VLMs.

- Extensive experimental analyses (see Section 4.2) demonstrate that our method, NegLabel, achieves state-of-the-art performance on multiple OOD detection benchmarks. Moreover, NegLabel demonstrates strong generalization capabilities across various VLM architectures. NegLabel also shows remarkable robustness in the face of diverse domain shifts.

## 2 PRELIMINARIES

Let $\mathcal{X}$ and $\mathcal{Y} = \{y_1, ..., y_K\}$ be the image space and ID label space, respectively. Note that $\mathcal{Y}$ is a set containing words, e.g., $\mathcal{Y} = \{\text{cat}, \text{dog}, \cdots, \text{bird}\}$, and $K$ is the number of ID classes. Given the ID feature random variable $X^{\text{in}} \in \mathcal{X}$ and OOD feature random variable $X^{\text{out}} \in \mathcal{X}$, we use the $\mathbb{P}_{X^{\text{in}}}$ and $\mathbb{P}_{X^{\text{out}}}$ to present the ID marginal distribution and OOD marginal distribution, respectively.

**CLIP-like models (Radford et al., 2021).** Given any input image $\mathbf{x} \sim \mathbb{P}_{X^{\text{in}}}$ and any label $y_j \in \mathcal{Y}$, we extract features of $\mathbf{x}$ and $y_j$ using the CLIP-like model $\mathbf{f}$, which consists of image encoder $\mathbf{f}^{\text{img}}$

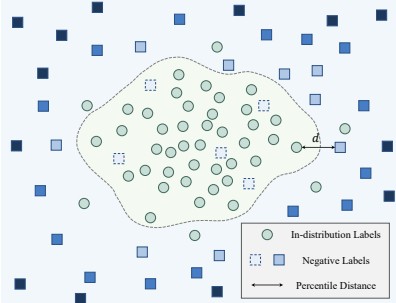

**Algorithm 1:** NegMining

> **Input** : Candidate labels $\mathcal{Y}^{\text{c}}$, ID labels $\mathcal{Y}$,
> Text encoder $\mathbf{f}^{\text{text}}$
> **Output**: Negative labels $\mathcal{Y}^{-}$
> // Calculate text embeddings
> 1 **for** $y_i \in \mathcal{Y}$ **do**
> 2  $\quad e_i = \mathbf{f}^{\text{text}}(\text{prompt}(y_i));$
>
> 3 **for** $\widetilde{y}_i \in \mathcal{Y}^{\text{c}}$ **do**
> 4  $\quad \widetilde{e}_i = \mathbf{f}^{\text{text}}(\text{prompt}(\widetilde{y}_i));$
> $\quad$ // Measure candidate-ID
> $\qquad$ label distance.
> 5  $\quad d_i = \text{percentile}_\eta(\{-\cos(\widetilde{e}_i, e_k)\}_{k=1}^K);$
> $\quad$ // Choose $M$ negative labels
> $\qquad$ from top-k distances.
> 6 $\mathcal{Y}^{-} = \text{topk}([d_1, d_2, \cdots, d_C], \mathcal{Y}^{\text{c}}, M).$

Figure 2: Illustration of NegMining. The algorithm selects negative labels with larger distances (lower similarities) from the ID labels. Darker blue squares represent the higher priorities to be picked. Dashed squares represent negative labels that are impossible to be selected.

and text encoder $\mathbf{f}^{\text{text}}$. Then the extracted features $\boldsymbol{h} \in \mathbb{R}^D$ and $\boldsymbol{e}_j \in \mathbb{R}^D$ are

$$\boldsymbol{h} = \mathbf{f}^{\text{img}}(\mathbf{x}), \quad \boldsymbol{e}_j = \mathbf{f}^{\text{text}}(\text{prompt}(y_j)), \quad \forall j = 1, 2, ..., K, \tag{1}$$

where $\text{prompt}(\cdot)$ represents the prompt template for input labels, and $D$ is the embedding dimension. In zero-shot classification, the goal is to predict the correct label or class for an image without prior training on that specific class. CLIP-like models' predictions are based on the cosine similarity of the image embedding features $\boldsymbol{h}$ and text embedding features $\boldsymbol{e}_1, \boldsymbol{e}_2, \cdots, \boldsymbol{e}_K$:

$$\hat{y} = \underset{y_j \in \mathcal{Y}}{\arg\max}\{\cos(\boldsymbol{h}, \boldsymbol{e}_j)\}, \quad \text{where } \boldsymbol{e}_j = \mathbf{f}^{\text{text}}(\text{prompt}(y_j)). \tag{2}$$

**Score functions.** Following many representative OOD detection methods (Huang et al., 2021; Liang et al., 2018; Liu et al., 2020), given a threshold $\gamma$ and a score function $S$, then $\mathbf{x}$ is detected as ID data if and only if $S(\mathbf{x}) \geq \gamma$:

$$G_\gamma(\mathbf{x}) = \text{ID, if } S(\mathbf{x}) \geq \gamma; \text{ otherwise, } G_\gamma(\mathbf{x}) = \text{OOD}. \tag{3}$$

The performance of OOD detection depends on how to design a score function $S$ to make OOD samples obtain lower scores while ID samples have higher scores.

**Problem Setting.** Zero-shot image classification for a pre-trained CLIP-like model is to classify images into one of given label names (i.e., ID classes), without any fine-tuning on ID classes (Radford et al., 2021). In this paper, we work with a pre-trained CLIP-like model and a specific set of class labels or names for a zero-shot classification task. Therefore, the primary objective of VLM-based OOD detection in our paper is to identify samples that do not belong to any known ID classes as OOD without sacrificing ID classification accuracy.

## 3 METHODOLOGY

As mentioned in Section 2, the semantic labels of ID samples are confined within the bounds of $\mathcal{Y}$, whereas OOD labels lie outside this label space. In contrast to traditional image classification tasks, zero-shot classifiers are trained on more extensive datasets (Radford et al., 2021; Jia et al., 2021) to gain a broader understanding of the training data. However, when conducting zero-shot image classification, the ID label space $\mathcal{Y}$ can weaken the model's ability to detect OOD samples, since the limited label space may not fully capture the differences between ID and OOD samples, ultimately reducing the model's sensitivity.

As a zero-shot classifier, a CLIP-like model can scale its label space using any text prompts with minimal overhead in cosine similarity calculations. Thus, the central concept behind NegLabel is to enlarge the model's zero-shot generalization ability by expanding the label space. By capitalizing on the scalability of the model's label space, we can effectively utilize the VLMs' text comprehension capabilities for zero-shot OOD detection.

### 3.1 SELECTION OF NEGATIVE LABELS

If there are labels $y^- \notin \mathcal{Y}$ that have lower affinities with ID images compared to OOD images, they can provide hints for detecting OOD samples. These labels are referred to as negative labels, denoted by $y^- \in \mathcal{Y}^-$, and the objective of selecting negative labels is to increase the gap between $\mathcal{Y}$ and $\mathcal{Y}^-$.

The large-scale corpus database contains almost all common concepts and entities worldwide. Its semantic space can effectively exploit the capacity of CLIP-like zero-shot models, allowing all input samples to be mapped to richer labels. As shown in Fig. 2, when selecting negative labels from the corpus, words with high similarity to the ID label space $\mathcal{Y}$ should be excluded so that the selected negative labels have lower affinities with ID samples than OOD samples. More analysis can be found in Appendix B.4. Then, we propose the NegMining algorithm to select suitable negative labels.

Given a large group of words [1] (contains tens of thousands of nouns, adjectives, etc.) fetched from lexical database like WordNet (Fellbaum, 1998), we can construct a candidate label space $\mathcal{Y}^c = \{\widetilde{y}_1, \widetilde{y}_2, ..., \widetilde{y}_C\}$, where $C$ is the total number of the candidate words. For each $\widetilde{y}_i(i = 1, 2, ..., C)$, we calculate the negative cosine similarities as the distance measure between the candidate label and the whole ID label space:

$$d_i = \text{percentile}_\eta(\{d_{ik}\}_{k=1}^K) = \text{percentile}_\eta(\{-\cos(\widetilde{e}_i, e_k)\}_{k=1}^K), \tag{4}$$

where $\widetilde{e}_i = \mathbf{f}^{\text{text}}(\text{prompt}(\widetilde{y}_i))$ is the text embedding of the candidate word $\widetilde{y}_i$, the $K$ is the number of ID classes, and $\text{percentile}_\eta, \eta \in [0, 1]$ represents the $100\eta$-th percentile of the data. When $\eta = 0$ means the minimum distance, we usually choose $\eta = 0.05$. We use the percentile of the distance between the candidate label $\widetilde{y}_i$ and all ID labels $\mathcal{Y}$ instead of the minimum distance as the metric to obtain robustness against outlier ID labels, as shown in Fig. 2.

Then we design the negative label selection criteria as $\mathcal{Y}^- = \text{topk}(\{d_1, d_2, \cdots, d_C\}, \mathcal{Y}^c, M)$. The operator $\text{topk}(A, B, M)$ first identifies the indices of the top-$M$ elements in set $A$, and then selects the corresponding elements from set $B$ based on these indices. The details are shown in Algorithm 1. In this work, unless noted otherwise, we use $M = 10000$ as the typical number of negative labels, which can achieve robust performance on various benchmarks.

It seems that it is better to directly select the negative labels far away from the ID image embeddings. However, in the zero-shot setting, the ID marginal distribution $\mathbb{P}_{X^{\text{in}}}$ is unreachable, i.e., there is no training data to guide the selection of negative labels. Fortunately, the CLIP-like model minimizes the embedding feature distance between paired image-text in a contrastive learning framework (Radford et al., 2021). Hence, we use ID labels to search for OOD concepts as negative labels that are far away from ID labels.

## 3.2 OOD Detection with Negative Labels

**NegLabel Score.** Given the selected negative label set $\mathcal{Y}^-$, we merge them with the ID labels $\mathcal{Y}$ to obtain an extended label space $\mathcal{Y}^{\text{ext}} = \mathcal{Y} \cup \mathcal{Y}^-$. Then, the extended labels are fed into the text encoder of CLIP to obtain text embeddings . CLIP calculates the cosine similarities between the text and image embeddings. The proposed OOD score can be formulated as

$$S(\mathbf{x}) = S^*(\text{sim}(\mathbf{x}, \mathcal{Y}), \text{sim}(\mathbf{x}, \mathcal{Y}^-)), \tag{5}$$

where $S^*(\cdot, \cdot)$ represents a fusion function that combines the similarity of a sample with ID labels $\text{sim}(\mathbf{x}, \mathcal{Y})$ and the similarity of the sample with negative labels $\text{sim}(\mathbf{x}, \mathcal{Y}^-)$.

As analyzed in Section 3.1, the selected negative labels are far from $\mathcal{Y}$, so ID samples have lower similarity with negative labels than those of OOD samples. Therefore, the design criterion of $S^*$ can be summarized as follows: $S^*$ is positively correlated with the similarity of the sample in the ID label space and negatively correlated with the similarity of the sample in the negative label space.

In the extended label space $\mathcal{Y}^{\text{ext}}$, the OOD detection task can be considered as the model's confidence that the image belongs to $\mathcal{Y}$. A natural design is to use a normalization function to check the proportion of the similarity $\text{sim}(\mathbf{x}, \mathcal{Y})$ in the extended label space $\mathcal{Y}^{\text{ext}}$. Here, we propose a NegLabel score in a sum-softmax form as follows, and more designs are discussed in Appendix A.5:

$$S(\mathbf{x}) = \sum_{i=1}^K e^{\cos(h, e_i)/\tau} / \left( \sum_{i=1}^K e^{\cos(h, e_i)/\tau} + \sum_{j=1}^M e^{\cos(h, \widetilde{e}_j)/\tau} \right), \tag{6}$$

where $K$ is the number of ID labels, $\tau$ is the temperature coefficient of the softmax function, and $M$ is the number of negative labels. Based on the analysis in Section 3.1, the affinities of the ID

---

[1]For the lexical database, we only retained nouns and adjectives as candidate labels because most of the other words either have their corresponding nouns or adjectives, or do not represent specific concepts or entities.

images to the negative labels are lower than those of the OOD images. Therefore, the ID samples will generate higher similarities on the ID labels, resulting in higher OOD scores.

Eq. (6) provides two special considerations for CLIP-like models:

Firstly, the cosine function clamps the similarity in $[-1, 1]$ and results in even smaller divergences between positive and negative pairs (e.g., the cosine similarities of positive pairs in CLIP are typically around 0.3 and that of negative pairs are around 0.1 (Ming et al., 2022b)). Such small differences may not adequately reflect the confidence variance of ID and OOD samples. Therefore, a temperature coefficient $\tau$ is introduced to scale the similarity, reflecting the model's confidence better.

Secondly, zero-shot models are not finetuned on task-specific ID data. Their classification performance is typically weaker than the fully supervised models, making it more susceptible to super-class confusion (e.g., mistaking a Husky for an Alaskan Malamute). The numerator of Eq. (6) uses the sum of the sample's similarity to all ID labels, making the OOD score more robust to super-class confusion. This design also improves the robustness of the zero-shot OOD detector to domain shifts, which will be analyzed in Section 4.2.

**Grouping Strategy.** In Section 3.1, we introduce the NegMining algorithm, which involves picking words that are as far away from the ID label as possible to expand the label space and avoid high similarities between ID images and negative labels (called false positives). Nevertheless, false positives may occur in the ID sample due to limited model performances and semantic overlaps between ID images and negative labels. This risk increases when more negative labels are used, increasing the variance of the OOD score. To balance the trade-off between the additional knowledge gained from negative labels and the risk of false positives, we propose a grouping strategy to divide negative labels into multiple groups and calculate the OOD score as Eq. (6) in each group separately, and then obtain the average OOD score across groups.

Specifically, we start by evenly dividing the selected $M$ negative labels into $n_g$ groups. Each group contains $\lfloor M/n_g \rfloor$ negative labels, and any remaining negative labels are discarded. We then calculate a separate NegLabel score for each group. Finally, we sum up the scores of all groups and take the average as the final NegLabel score. The grouping strategy limits the risk of false positives within each group, and further smoothes this noise through averaging across groups. More detailed analysis can be found in Appendix A.5.

### 3.3 UNDERSTANDING NEGATIVE LABELS FROM THE PERSPECTIVE OF MULTI-LABEL CLASSIFICATION

To better understand how negative labels can facilitate zero-shot OOD detection, we provide theoretical analysis from the perspective of multi-label classification to demonstrate that negative labels can improve the separability of ID and OOD samples.

For VLMs like CLIP, whether an image and a label match is determined by calculating the similarities between their embeddings. To simplify the analysis, we assume that the selected negative labels, $\widetilde{y}_i \in \mathcal{Y}^-$, are independently and identically distributed (we will discuss the validity of this assumption later). For each $\widetilde{y}_i \in \mathcal{Y}^-, i = 1, 2, ..., M$, the input image $\mathbf{x}$ has an implicit distribution noted as $H(s_i; f, \mathbf{x}, \widetilde{y}_i)$ for its similarity score $s_i = \text{sim}(\mathbf{x}, \widetilde{y}_i)$. By applying a threshold $\psi$, the similarity score can be converted into a binary label $s_i^* = \mathbb{1}_{[s_i \geq \psi]}$. Thus, we can define $p = P\left(s_i \geq \psi | f, \mathbf{x}, \widetilde{y}_i\right)$ as the probability of classifying the image as positive for the given label $\widetilde{y}_i$. The introduced negative labels can be regarded as an $M$-way multi-label classification task on $\mathcal{Y}^-$.

Under the multi-label classification setting, for each negative label $\widetilde{y}_i, i = 1, 2, ..., M$, the single-label classification result of an input image $\mathbf{x}$ follows the Bernoulli distribution with a parameter $p$, where $p$ is the probability that the label of $\mathbf{x}$ is $\widetilde{y}_i$. Hence, based on the relation between Bernoulli and binomial distributions, for the input image $\mathbf{x}$, the positive count $c = \sum_i s_i^*$ among $M$ negative labels follows the binomial distribution $B(M, p)$[2]. Specifically, we define the in-distribution positive count

---

[2]However, due to the large number of negative labels, covering a wide semantic space, their affinity with the sample $x$ is variable. Thus, we provide a more general case based on a Poisson binomial distribution in Appendix B.6. Specifically, we assume that the probability of an image $x$ matching with a negative label is $p_i$, where different negative labels have different probabilities. This better reflects the real-world scenario, and thus, we consider the number of matches between an image $x$ and them to follow a Poisson binomial distribution.

as $c^{\text{in}} \sim B(M, p_1), \mathbf{x} \sim \mathbb{P}_{X^{\text{in}}}$, and the out-of-distribution positive count as $c^{\text{out}} \sim B(M, p_2), \mathbf{x} \sim \mathbb{P}_{X^{\text{out}}}$. Recall that the affinities between ID images and negative labels are lower than that of OOD images, so we have $p_1 < p_2$ holds.

Next, we consider a toy OOD score function that only takes the positive count as criteria, i.e., $\widetilde{S}(\mathbf{x}) = -c$, which is a special case of Eq. (5). Then we have $\widetilde{S}^{\text{in}} \sim B(M, p_1)$ and $\widetilde{S}^{\text{out}} \sim B(M, p_2)$. Because the number of negative labels is sufficiently large, according to the binomial approximation rules [3], the OOD scores $\widetilde{S}^{\text{in}}$ and $\widetilde{S}^{\text{out}}$ can be regarded as following the normal distribution $\mathcal{N}(Mp_1, Mp_1(1 - p_1))$ and $\mathcal{N}(Mp_2, Mp_2(1 - p_2))$, respectively.

We use $\text{FPR}_\lambda$ as the performance metric of an OOD detector to show the separability between the ID and OOD samples. The $\text{FPR}_\lambda$ metric in relation to the toy function $\widetilde{S}$ can be defined as follows:

$$\text{FPR}_\lambda = \Phi\left[\Phi^{-1}\left[\lambda; Mp_1, \sqrt{Mp_1(1 - p_1)}\right]; Mp_2, \sqrt{Mp_2(1 - p_2)}\right]$$
$$= \frac{1}{2} + \frac{1}{2} \cdot \text{erf}\left(\sqrt{\frac{p_1(1 - p_1)}{p_2(1 - p_2)}} \text{erf}^{-1}(2\lambda - 1) + \frac{\sqrt{M}(p_1 - p_2)}{\sqrt{2p_2(1 - p_2)}}\right), \tag{7}$$

where $\Phi[\cdot; \mu, \sigma]$ represent the CDF of a normal distribution with $\mu$ as mean and $\sigma$ as standard deviation, $\text{erf}(x) = \frac{2}{\sqrt{\pi}} \int_0^x e^{-t^2} \, dt$ represents the error function.

We target investigating the relationship between the $\text{FPR}_\lambda$ metric and the number of negative labels $M$. Therefore, we calculate the partial derivative with respect to $M$ and use $z = \sqrt{\frac{p_1(1-p_1)}{p_2(1-p_2)}} \text{erf}^{-1}(2\lambda - 1) + \frac{\sqrt{M}(p_1-p_2)}{\sqrt{2p_2(1-p_2)}}$ to simplify the expression:

$$\frac{\partial \text{FPR}_\lambda}{\partial M} = \frac{1}{2} \cdot \frac{\partial \text{erf}(z)}{\partial z} \cdot \frac{\partial z}{\partial M} = \frac{e^{-z^2}}{2\sqrt{2\pi}} \cdot \frac{p_1 - p_2}{\sqrt{Mp_2(1 - p_2)}} < 0. \tag{8}$$

It is evident that as $M$ increases, $\text{FPR}_\lambda$ consistently decreases. This observation suggests that the performance of the OOD detector is enhanced by incorporating more negative labels (note that $p_1 - p_2 < 0$). Consequently, the inclusion of negative labels provides additional information that aids in distinguishing between ID and OOD samples. Furthermore, Eq. (8) is negatively correlated with $p_2 - p_1$, i.e., the difference between the probability that OOD and ID samples are positively classified w.r.t. the negative labels. This indicates that selecting negative labels with significant differences in similarity with ID and OOD samples can help improve OOD detection performance, and it explains the role of the NegMining algorithm.

Finally, we discuss the plausibility of this theoretical modeling. Readers may notice that as $M$ increases, $z$ approaches $+\infty$ and $\text{FPR}_\lambda$ approaches 0. However, this does not mean that a large number of negative labels can infinitely optimize the performance of the OOD detector, as the semantic space formed by images is limited. With $M$ increases, the newly added negative labels either become less differentiated on ID and OOD images or exhibit semantic overlap with the previous label. Consequently, this leads to a decline in the difference between the probabilities $p_2$ and $p_1$, or in some cases, these additional labels may no longer satisfy the assumption of independent and identically distributed data. In particular, if the similarities between negative labels and ID images are larger (i.e., $p_1 > p_2$), it will cause an increase in $\text{FPR}_\lambda$. The detailed derivations of Eqs. (7) and (8) can be found in Appendix B.5 and more theoretical analysis can be found in Appendix B.6.

## 4 EXPERIMENTS

### 4.1 EXPERIMENT SETUP

**Datasets and benchmarks.** We evaluate our method on the ImageNet-1k OOD benchmark (Huang et al., 2021) and compare it with various previous methods. The ImageNet-1k OOD benchmark is a widely used performance validation method that uses the large-scale visual dataset ImageNet-1k as ID data and iNaturalist (Horn et al., 2018), SUN (Xiao et al., 2010), Places (Zhou et al., 2018), and Textures (Cimpoi et al., 2014) as OOD data, covering a diverse range of scenes and semantics. We

---

[3]If $Mp \geq 5$ and $M(1 - p) \geq 5$, the binomial distribution can be approximated by a Gaussian distribution (Parameswaran, 1979).

Table 1: OOD detection performance comparison with baselines. All methods are based on CLIP-B/16. The ID data are ImageNet-1k. All values are percentages. ↑ indicates larger values are better and ↓ indicates smaller values are better. The shadow part represents our method.

| Methods | OOD Dataset | | | | | | | | | |
| | iNaturalist | | SUN | | Places | | Textures | | Average | |
| | AUROC↑ | FPR95↓ | AUROC↑ | FPR95↓ | AUROC↑ | FPR95↓ | AUROC↑ | FPR95↓ | AUROC↑ | FPR95↓ |
|---|---|---|---|---|---|---|---|---|---|---|
| **Requires training (or w. fine-tuning)** | | | | | | | | | | |
| MSP (Hendrycks & Gimpel, 2017) | 87.44 | 58.36 | 79.73 | 73.72 | 79.67 | 74.41 | 79.69 | 71.93 | 81.63 | 69.61 |
| ODIN (Liang et al., 2018) | 94.65 | 30.22 | 87.17 | 54.04 | 85.54 | 55.06 | 87.85 | 51.67 | 88.80 | 47.75 |
| Energy (Liu et al., 2020) | 95.33 | 26.12 | 92.66 | 35.97 | 91.41 | 39.87 | 86.76 | 57.61 | 91.54 | 39.89 |
| GradNorm (Huang et al., 2021) | 72.56 | 81.50 | 72.86 | 82.00 | 73.70 | 80.41 | 70.26 | 79.36 | 72.35 | 80.82 |
| ViM (Wang et al., 2022) | 93.16 | 32.19 | 87.19 | 54.01 | 83.75 | 60.67 | 87.18 | 53.94 | 87.82 | 50.20 |
| KNN (Sun et al., 2022) | 94.52 | 29.17 | 92.67 | 35.62 | 91.02 | 39.61 | 85.67 | 64.35 | 90.97 | 42.19 |
| VOS (Du et al., 2022) | 94.62 | 28.99 | 92.57 | 36.88 | 91.23 | 38.39 | 86.33 | 61.02 | 91.19 | 41.32 |
| NPOS (Tao et al., 2023) | 96.19 | 16.58 | 90.44 | 43.77 | 89.44 | 45.27 | 88.80 | 46.12 | 91.22 | 37.93 |
| **Zero-shot (no training required)** | | | | | | | | | | |
| Mahalanobis (Lee et al., 2018) | 55.89 | 99.33 | 59.94 | 99.41 | 65.96 | 98.54 | 64.23 | 98.46 | 61.50 | 98.94 |
| Energy (Liu et al., 2020) | 85.09 | 81.08 | 84.24 | 79.02 | 83.38 | 75.08 | 65.56 | 93.65 | 79.57 | 82.21 |
| ZOC (Esmaeilpour et al., 2022) | 86.09 | 87.30 | 81.20 | 81.51 | 83.39 | 73.06 | 76.46 | 98.90 | 81.79 | 85.19 |
| MCM (Ming et al., 2022a) | 94.59 | 32.20 | 92.25 | 38.80 | 90.31 | 46.20 | 86.12 | 58.50 | 90.82 | 43.93 |
| CLIPN (Wang et al., 2023) | 95.27 | 23.94 | 93.93 | 26.17 | **92.28** | **33.45** | **90.93** | **40.83** | 93.10 | 31.10 |
| NegLabel | **99.49** | **1.91** | **95.49** | **20.53** | 91.64 | 35.59 | 90.22 | 43.56 | **94.21** | **25.40** |

also follow the settings of MCM (Ming et al., 2022a) and use Stanford-Cars (Krause et al., 2013), CUB-200 (Wah et al., 2011), Oxford-Pet (Parkhi et al., 2012), Food-101 (Bossard et al., 2014), and some subsets of ImageNet-1k (Deng et al., 2009) as ID data, and iNaturalist, SUN, Places , and Textures as OOD data to conduct validation experiments on our method. MCM uses subsets of ImageNet-1k for fine-grained analysis. For example, MCM constructs ImageNet-10, which mimics the class distribution of CIFAR-10 (Krizhevsky, 2009) but has high-resolution images. Each OOD dataset has no classes that overlap with the ID dataset.

**Implement details.** In this paper, we implement NegLabel on various VLM architectures, including CLIP (Radford et al., 2021), ALIGN (Jia et al., 2021), GroupViT (Xu et al., 2022), AltCLIP (Chen et al., 2022). Unless otherwise specified, we employ CLIP-B/16 for zero-shot OOD detection, and the hyperparameters use the following default configurations. The NegMining algorithm takes WordNet as the corpus and selects $M = 10000$ negative labels under $\eta = 0.05$. We use the NegLabel score in the sum-softmax form and set $\tau = 0.01$ as the temperature, and use $n_g = 100$ for grouping strategy by default. More experiment settings can be found in Appendix A.1.

**Computational cost.** NegLabel is a post hoc OOD detector with about $O(2MD)$ FLOPs extra computational burden per image, which usually introduce $< 1\%$ network forward latency. When taking $M = 10000$ negative labels, it takes $\sim$1ms per sample for OOD detection. NegLabel detects OOD samples in parallel with the zero-shot classification task without any impact on ID classification performance. The NegMining algorithm selects and dumps the negative label before the inference phase, and it usually takes a few minutes to process a large-scale corpus on a single 3090Ti GPU.

## 4.2 EXPERIMENTAL RESULTS AND ANALYSIS

**OOD detection performance comparison with baselines.** We compare our proposed method, NegLabel, with other existing OOD detection methods in Table 1. The methods we compared with include MCM, a method specifically designed for zero-shot OOD detection, as well as traditional methods that were re-implemented using a finetuned CLIP on the ImageNet-1k, see Appendix A.1 for more details. Our method NegLabel achieves the state-of-the-art on the ImageNet-1k benchmark, which highlights its superior performance in the zero-shot setting. Furthermore, our method can surpass traditional methods with a finetuned CLIP, demonstrating CLIP's strong OOD detection capabilities in zero-shot scenarios. This is because CLIP can parse images in a fine-grained manner, which is achieved through its pre-training on a large-scale image-text dataset. Additionally, our choice of negative labels provides more semantic information, which further enhances the separability of ID and OOD samples. It is worth noting that our proposed method achieved surprisingly high performance on the iNaturalist dataset, and we analyze the reason in Appendix B.1. More discussions about ZOC and CLIPN are in Appendix B.2 and Appendix B.3. We also follow MCM to conduct experiments on the small-scale benchmark in Appendix A.2. The further investigations with some case studies can be found in Appendix B.4.

**Zero-shot OOD detection performance comparison on hard OOD detection.** Following MCM's setup, we also explore the performance of NegLabel on hard OOD detection tasks, as shown in Table 2. For semantically hard OOD detection, we alternate using ImageNet-10 and ImageNet-20 as

Table 2: Zero-shot OOD detection performance comparison on hard OOD detection tasks.

| ID Dataset | OOD Dataset | Method | AUROC↑ | FPR95↓ |
|---|---|---|---|---|
| ImageNet-10 | ImageNet-20 | MCM | 98.60 | 6.00 |
| | | NegLabel | **98.86** | **5.10** |
| ImageNet-20 | ImageNet-10 | MCM | 98.09 | 13.04 |
| | | NegLabel | **98.81** | **4.60** |
| ImageNet-10 | ImageNet-100 | MCM | 99.39 | 2.50 |
| | | NegLabel | **99.51** | **1.68** |
| ImageNet-100 | ImageNet-10 | MCM | 87.20 | 60.00 |
| | | NegLabel | **90.19** | **40.20** |
| WaterBirds | Spurious OOD | MCM | 93.30 | 14.45 |
| | | NegLabel | **94.67** | **9.50** |

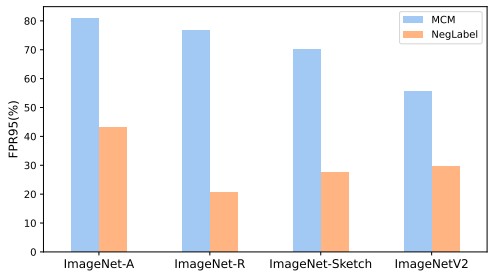

Figure 3: Zero-shot OOD detection performance robustness to domain shift.

ID and OOD data, as well as using ImageNet-10 and ImageNet-100 to mimic the setting proposed by (Fort et al., 2021) with high-resolution images. The results in Table 2 show that NegLabel can significantly outperform MCM in all four experimental settings, demonstrating that NegLabel has strong discriminative power for semantically hard OOD data. For spurious OOD detection, since MCM does not provide specific parameters for generating spurious OOD samples, we reproduce one setting from Ming et al. (2022b) for experimental comparison. The last row of the experimental data also shows that NegLabel can surpass MCM in zero-shot spurious OOD detection tasks.

**Robustness to domain shift.** To test the domain generalization capability, we use several versions of ImageNet with diverse domain shifts as ID samples. OOD samples are still taken from iNaturalist, SUN, Places, and Textures. The experimental details can be found in Appendix A.3 and the average results on four OOD datasets are shown in Fig. 3. It is apparent that the zero-shot OOD detection performance of MCM significantly deteriorates across diverse domain shifts, indicating the difficulty of zero-shot OOD detection under such conditions. Nevertheless, our method NegLabel continues to achieve noteworthy performances cross diverse ID datasets, thus demonstrating its remarkable robustness against diverse domain shifts. MCM heavily relies on the model's classification ability on the ID dataset. Therefore, when a domain shift leads to a decrease in ID classification accuracy, the OOD detection performance of MCM also decreases accordingly. In contrast, our approach NegLabel treats the ID labels $\mathcal{Y}$ as a whole, which is less affected by inter-class confusion, making it more robust to domain shifts.

**Ablation on each module in NegLabel.** We conduct an ablation analysis on each module in our proposed zero-shot OOD detection framework, as shown in Table 3. Note that when NegMining is not used, negative labels are randomly selected from the corpus.

Comparing the second and fourth rows, negative labels selected by NegMining show +1.81% AUROC and -6.63% FPR95, indicating that negative labels require greater semantic differences from ID labels. Comparing the second and third rows, the grouping strategy only provides a slight improvement when NegMining is not used. However, comparing the first and fourth rows, it yields +0.45% AUROC and -2.79% FPR95 with selected negative labels, further demonstrating that the grouping strategy aims to reduce coincidental false positives between ID images and negative labels. The first row and the last three rows demonstrate that an increasing number of negative labels gradually improves the performance of OOD detection and then declines. This indicates that NegMining prioritizes selecting negative labels that are farther away from the ID labels, as they are more discriminative for detecting OOD images. More detailed analyses can be found in Appendices A.4 and A.5.

**Ablation on VLM architectures.** We evaluate NegLabel with different VLM architectures and the results are shown in Table 4. The first part shows results with different backbones of CLIP, illustrating that the larger backbone boosts the performance of OOD detection. The second part shows NegLabel can generalize well to various VLM architectures. It is important to note that we take the same hyperparameters across all architectures, demonstrating the robustness of NegLabel's hyperparameters on different VLM architectures.

## 5 RELATED WORKS

**Traditional OOD detection.** There exist two major categories of visual OOD detection: training-time regularization (Hsu et al., 2020a; Bitterwolf et al., 2020; Wang et al., 2021b; Bitterwolf et al., 2020; Liu et al., 2019; Vaze et al., 2022; Yang et al., 2021) and post hoc methods (Liang et al., 2018; Liu et al., 2020; Hendrycks & Gimpel, 2017; Lee et al., 2018; Wang et al., 2021a; Jiang et al., 2023).

Table 3: Module ablation in NegLabel. All methods are based on CLIP-B/16. The ID data are ImageNet-1k. More can be found in Appendices A.4 and A.5.

| Number of Negative Labels | NegMining | Grouping Strategy | AUROC | FPR95 |
|---|---|---|---|---|
| 10,000 | ✓ | ✓ | **94.21** | **25.40** |
| 10,000 | ✗ | ✗ | 91.95 | 34.82 |
| 10,000 | ✗ | ✓ | 92.01 | 34.69 |
| 10,000 | ✓ | ✗ | 93.76 | 28.19 |
| 1,000 | ✓ | ✓ | 91.60 | 36.14 |
| 20,000 | ✓ | ✓ | 93.85 | 26.56 |
| 50,000 | ✓ | ✓ | 92.70 | 31.67 |

Table 4: Comparison with different VLM architectures on ImageNet-1k (ID). Detailed results can be found in Appendix A.6.

| Architecture | Backbone | AUROC↑ | FPR95↓ |
|---|---|---|---|
| CLIP | ResNet50 | 92.97 | 30.70 |
| | ResNet50x16 | 93.43 | 29.50 |
| | ViT-B/32 | 93.67 | 27.92 |
| | ViT-B/16 | 94.21 | 25.40 |
| | ViT-L/14 | 94.47 | 24.81 |
| ALIGN | EfficientNet-B7 | 90.76 | 39.66 |
| AltCLIP | ViT-L/14 | 94.60 | 23.63 |
| GroupViT | GroupViT | 91.72 | 33.10 |

Training-time regularization methods assume that a portion of OOD data can be accessed during model training. ConfBranch (DeVries & Taylor, 2018) constructs an additional branch from the penultimate layer of the classifier to learn the ID/OOD discrimination task. The confidence output from this branch serves as the OOD score during the inference time. G-ODIN (Hsu et al., 2020b) decomposes prior knowledge to model the probability of ID-ness. CSI (Tack et al., 2020) optimizes the OOD detector using contrastive learning. MOS (Huang et al., 2021) groups all categories in advance and adds an additional class to each group to redesign the loss for training. VOS (Du et al., 2022) generates better energy scores by synthesizing virtual anomalies. LogitNorm (Wei et al., 2022) provides an alternative loss for cross-entropy loss that separates the influence of logit norm from the training process. CIDER (Ming et al., 2023) improves OOD detection performance by optimizing contrastive loss based on the distance between class prototypes and the distance between samples and class prototypes on the basis of KNN (Sun et al., 2022).

Post hoc methods do not change the model's parameters and are typically accomplished by designing the OOD score. MSP (Hendrycks & Gimpel, 2017) uses the maximum predicted softmax probability as the OOD score. ODIN (Liang et al., 2018) enhances MSP by perturbing the inputs and rescaling the logits. Energy (Liu et al., 2020) proposes to use the energy function (LeCun et al., 2006) for measuring OOD-ness. ReAct (Sun et al., 2021) further improves the performance of the Energy score by feature clipping, while DICE (Sun & Li, 2022) improves the performance of the Energy score by discarding most salient weights in the fully connected layer. Mahalanobis (Lee et al., 2018) computes the minimum Mahalanobis distance between the feature and the class-wise centroids as the OOD score. GradNorm (Huang et al., 2021) designs the OOD score in the gradient space. Lastly, ViM (Wang et al., 2022) combines the norm of feature residual with the principal space formed by training features and the original logits to compute the degree of OOD-ness. KNN (Sun et al., 2022) investigates the effectiveness of non-parametric nearest-neighbor distance for detecting OOD samples. ASH (Djurisic et al., 2023) works by pruning a large portion of an input sample's activation and lightly adjusting the remaining.

**CLIP-based OOD detection.** The concept of zero-shot outlier exposure, as defined by (Fort et al., 2021), assumes that the labels of OOD images are known and uses them as candidate labels to complete the outlier exposure task. Recently, there has been some research into the problem of zero-shot OOD detection. For instance, ZOC (Esmaeilpour et al., 2022) proposes using a trainable generator and additional data to generate candidate OOD labels. MCM (Ming et al., 2022a) is the earliest post hoc zero-shot OOD detection method that uses temperature scaling strategy and maximum predicted softmax value as the OOD score. Based on MCM, NPOS (Tao et al., 2023) finetunes CLIP's image encoder using generated OOD data to optimize the decision boundary between ID and OOD data for the visual OOD detection task. Our method, NegLabel, is a post hoc zero-shot OOD detection method that leverages CLIP's zero-shot capabilities for text to design OOD scores.

## 6 CONCLUSION

This paper proposes a simple yet effective post hoc zero-shot detection framework called NegLabel. The approach involves incorporating a large set of negative labels that exhibit significant semantic differences from the ID labels. The proposed method determines whether an image belongs to OOD by comparing its affinity towards ID and negative labels. To enhance the distinction between ID and OOD samples, the NegMining algorithm is proposed to select high-quality negative labels. Moreover, we design a new scheme for the zero-shot OOD score that effectively leverages the knowledge of VLMs. Experimental results show that the proposed method achieves state-of-the-art performance on multiple zero-shot OOD detection benchmarks and generalizes well on multiple VLM architectures. NegLabel also presents remarkable robustness against diverse domain shifts.

## ACKNOWLEDGMENTS

This work was supported by the National Natural Science Foundation of China (Grant NO. 62122035, and 62376104). XJ and BH were supported by the NSFC General Program No. 62376235, Guangdong Basic and Applied Basic Research Foundation No. 2022A1515011652, HKBU Faculty Niche Research Areas No. RC-FNRA-IG/22-23/SCI/04, and HKBU CSD Departmental Incentive Scheme. FL was supported by the Australian Research Council with grant numbers DP230101540 and DE240101089, and the NSF&CSIRO Responsible AI program with grant number 2303037. TLL was partially supported by the following Australian Research Council projects: FT220100318, DP220102121, LP220100527, LP220200949, and IC190100031.

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

# A  FURTHER EXPERIMENTS

## A.1  DETAILED EXPERIMENT SETTING.

### A.1.1  IMPLEMENT DETAILS.

The proposed method NegLabel is implemented with Python 3.9 and PyTorch 1.9. All the experiments on a single NVIDIA GeForce RTX 3090Ti GPU. NegLabel is a post hoc OOD detector with about $\mathcal{O}(2MD)$ FLOPs extra computational burden per image (note that $M$ is the number of negative labels and $D$ is the dimension of the embedding feature), which usually introduce $< 1\%$ network forward latency. NegLabel detects OOD samples in parallel with the zero-shot classification task, without any impact on ID classification performance. The NegMining algorithm selects and dumps the negative label in advance, and it usually takes a few minutes for processing the large-scale corpus like WordNet on a single 3090Ti GPU.

In this paper, we implement NegLabel on various VLM architectures, including CLIP (Radford et al., 2021), ALIGN (Jia et al., 2021), GroupViT (Xu et al., 2022), AltCLIP (Chen et al., 2022). Unless otherwise specified, we employ CLIP-B/16 for zero-shot OOD detection and the hyperparameters use the following default configurations. The NegMining algorithm takes WordNet as the corpus and select $M = 10000$ negative labels under $\eta = 0.05$. We use the NegLabel score in the sum-softmax form and set $\tau = 0.01$ as the temperature, and use $n_g = 100$ for grouping strategy by default.

Table 1 reports the results of various mathods that perform OOD detection on fine-tuned CLIP. We follow the fine-tuning strategy of NPOS (Tao et al., 2023), which fine-tunes the last two blocks of CLIP's image encoder and equips a fully connected layer after the image encoder, which maps the image features into class predictions.

### A.1.2  EVALUATION METRICS.

We use two common metrics to evaluate OOD detection methods Huang et al. (2021): the false positive rate that OOD data are classified as ID data when 95% of ID data are correctly classified (FPR95) Provost et al. (1998) and the *area under the receiver operating characteristic curve* (AUROC) Huang et al. (2021).

## A.2  ZERO-SHOT OOD DETECTION PERFORMANCE COMPARISON ACROSS DIVERSE TASKS.

We compare the zero-shot OOD detection performance of NegLabel and MCM on seven different in-distribution (ID) datasets. As shown in Table 5, our method outperforms MCM on all OOD datasets regardless of the ID dataset used. Specifically, NegLabel achieved an AUROC of around 99.9% and FPR95 below 1% on Standford-Cars, CUB-200, Oxford-Pet, Food-101, and ImageNet-10, while MCM could only achieve this performance on Standford-Cars and ImageNet-10. This demonstrates the superior OOD detection performance of our method on these datasets. When ImageNet-100 is used as the ID dataset, our method was able to achieve an average improvement of approximately 4% AUROC and a decrease of around 20% FPR95 on four OOD datasets compared to MCM. Overall, our method showes strong zero-shot OOD detection capabilities on different ID datasets.

## A.3  ROBUSTNESS TO DOMAIN SHIFT.

To test the domain generalization capability, we use several versions of ImageNet with diverse domain shifts as ID samples. The ImageNet-Sketch dataset (Wang et al., 2019) consists of 50,000 images, with 50 images for each of the 1,000 ImageNet classes, rendered in a "black and white" color scheme. ImageNet-A (Hendrycks et al., 2021b) is a dataset of images labeled with ImageNet labels, collected by selecting only the images that ResNet-50 models fail to classify correctly. ImageNet-R (Hendrycks et al., 2021a) is a dataset that includes various artistic representations such as art, cartoons, deviantart, graffiti, embroidery, graphics, origami, paintings, patterns, plastic objects, plush objects, sculptures, sketches, tattoos, toys, and video game renditions of ImageNet classes. It consists of 30,000 images, representing 200 ImageNet classes. The ImageNetV2 dataset (Recht et al., 2019) includes new test data specifically created for the ImageNet benchmark. These test sets were sampled after a decade of progress on the original ImageNet dataset. OOD samples are still taken from iNaturalist, SUN, Places, and Textures. There is no overlapping classes between ID and OOD samples.

Table 5: Zero-shot OOD detection results based on CLIP-B/16 with various ID datasets.

| ID Dataset | Method | iNaturalist | | SUN | | Places | | Textures | | Average | |
|---|---|---|---|---|---|---|---|---|---|---|---|
| | | AUROC↑ | FPR95↓ | AUROC↑ | FPR95↓ | AUROC↑ | FPR95↓ | AUROC↑ | FPR95↓ | AUROC↑ | FPR95↓ |
| Stanford-Cars | MCM | 99.77 | 0.05 | 99.95 | 0.02 | 99.89 | 0.24 | 99.96 | 0.02 | 99.89 | 0.08 |
| | NegLabel | **99.99** | **0.01** | **99.99** | **0.01** | **99.99** | **0.03** | **99.99** | **0.01** | **99.99** | **0.01** |
| CUB-200 | MCM | 98.24 | 9.83 | 99.10 | 4.93 | 98.57 | 6.65 | 98.75 | 6.97 | 98.66 | 7.09 |
| | NegLabel | **99.96** | **0.18** | **99.99** | **0.02** | **99.90** | **0.33** | **99.99** | **0.01** | **99.96** | **0.13** |
| Oxford-Pet | MCM | 99.38 | 2.85 | 99.73 | 1.06 | 99.56 | 2.11 | 99.81 | 0.80 | 99.62 | 1.70 |
| | NegLabel | **99.99** | **0.01** | **99.99** | **0.02** | **99.96** | **0.17** | **99.97** | **0.11** | **99.98** | **0.07** |
| Food-101 | MCM | 99.78 | 0.64 | 99.75 | 0.90 | 99.58 | 1.86 | 98.62 | 4.04 | 99.43 | 1.86 |
| | NegLabel | **99.99** | **0.01** | **99.99** | **0.01** | **99.99** | **0.01** | **99.60** | **1.61** | **99.90** | **0.40** |
| ImageNet-10 | MCM | 99.80 | 0.12 | 99.79 | 0.29 | 99.62 | 0.88 | 99.90 | 0.04 | 99.78 | 0.33 |
| | NegLabel | **99.83** | **0.02** | **99.88** | **0.20** | **99.75** | **0.71** | **99.94** | **0.02** | **99.85** | **0.24** |
| ImageNet-20 | MCM | 99.66 | 1.02 | 99.50 | 2.55 | **99.11** | **4.40** | 99.03 | 2.43 | 99.32 | 2.60 |
| | NegLabel | **99.95** | **0.15** | **99.51** | **1.93** | 98.97 | **4.40** | **99.11** | **2.41** | **99.39** | **2.22** |
| ImageNet-100 | MCM | 96.77 | 18.13 | 94.54 | 36.45 | 94.36 | 34.52 | 92.25 | 41.22 | 94.48 | 32.58 |
| | NegLabel | **99.87** | **0.57** | **97.89** | **11.26** | **96.25** | **19.15** | **96.00** | **20.37** | **97.50** | **12.84** |

The detailed results shown in Table 6 illustrate that the zero-shot OOD detection performance of MCM significantly deteriorates across diverse domain shifts, indicating the difficulty of zero-shot OOD detection under such conditions. Nevertheless, our method NegLabel continues to achieve noteworthy performances cross diverse ID datasets, thus demonstrating its remarkable robustness against diverse domain shifts. MCM heavily relies on the model's classification ability on the ID dataset. Therefore, when a domain shift leads to a decrease in ID classification accuracy, the OOD detection performance of MCM also decreases accordingly. In contrast, our approach NegLabel treats the ID labels $\mathcal{Y}$ as a whole, which is less affected by inter-class confusion, making it more robust to domain shifts.

Table 6: Zero-shot OOD detection performance robustness to domain shift. All methods are based on CLIP-B/16. The ID data are ImageNet-Sketch. All values are percentages. ↑ indicates larger values are better and ↓ indicates smaller values are better. The shadow part represents our method.

| ID Dataset | Method | iNaturalist | | SUN | | Places | | Textures | | Average | |
|---|---|---|---|---|---|---|---|---|---|---|---|
| | | AUROC↑ | FPR95↓ | AUROC↑ | FPR95↓ | AUROC↑ | FPR95↓ | AUROC↑ | FPR95↓ | AUROC↑ | FPR95↓ |
| ImageNet-Sketch | MCM | 87.74 | 63.06 | 85.35 | 67.24 | 81.19 | 70.64 | 74.77 | 79.59 | 82.26 | 70.13 |
| | NegLabel | **99.34** | **2.24** | **94.93** | **22.73** | **90.78** | **38.62** | **89.29** | **46.10** | **93.59** | **27.42** |
| ImageNet-A | MCM | 79.50 | 76.85 | 76.19 | 79.78 | 70.95 | 80.51 | 61.98 | 86.37 | 72.16 | 80.88 |
| | NegLabel | **98.80** | **4.09** | **89.83** | **44.38** | **82.88** | **60.10** | **80.25** | **64.34** | **87.94** | **43.23** |
| ImageNet-R | MCM | 83.22 | 71.51 | 80.31 | 74.98 | 75.53 | 76.67 | 67.66 | 83.72 | 76.68 | 76.72 |
| | NegLabel | **99.58** | **1.60** | **96.03** | **15.77** | **91.97** | **29.48** | **90.60** | **35.67** | **94.54** | **20.63** |
| ImageNetV2 | MCM | 91.79 | 45.90 | 89.88 | 50.73 | 86.52 | 56.25 | 81.51 | 69.57 | 87.43 | 55.61 |
| | NegLabel | **99.40** | **2.47** | **94.46** | **25.69** | **90.00** | **42.03** | **88.46** | **48.90** | **93.08** | **29.77** |

## A.4 ABLATION STUDY ON THE NEGMINING ALGORITHM.

We conduct an ablation analysis on each module in our proposed zero-shot OOD detection framework and the detailed results are shown in Table 7.

The first row displays the performance of NegLabel, while the subsequent rows represent ablation studies on modules and the number of negative labels. Note that when NegMining is not used, negative labels are randomly selected from the corpus.

Comparing the second and fourth rows, negative labels selected by NegMining show +1.81% AUROC and -6.63% FPR95, indicating that negative labels require greater semantic differences from ID labels. Comparing the second and third rows, the grouping strategy only provides a slight improvement when NegMining is not used. However, comparing the first and fourth rows, it yields +0.45% AUROC and -2.79% FPR95 with selected negative labels, further demonstrating that the grouping strategy aims to reduce coincidental false positives between ID images and negative labels.

The first row and the last three rows of Table 7 demonstrate the impact of an increasing number of negative labels on the performance of OOD detection. The fifth row shows that even with a small number of negative labels, our method can still achieve state-of-the-art performance. Furthermore, there is a noticeable performance gain in OOD detection when the number of negative labels increases to 10,000. However, as the NegMining algorithm gives priority to candidate labels that are more dissimilar from the ID labels, when the number further increases to 20,000, the newly added labels exhibit more dissimilarities with the ID labels. Consequently, this leads to a decline in performance. This finding is consistent with the analysis presented in Section 3.3.

Interestingly, even when we indiscriminately use all the words in WordNet, the OOD detector still demonstrates remarkable performance and surpasses previous methods. This phenomenon suggests that the broad semantic space constructed by negative labels contributes to harnessing the latent knowledge of VLMs and provides new clues for OOD detection. For more detailed experiments regarding the number of negative labels, please refer to Appendix A.4.1.

Table 7: Module ablation in NegLabel. All methods are based on CLIP-B/16. The ID data are ImageNet-1k.

| Number of Negative Labels | NegMining | Grouping Strategy | OOD Dataset. | | | | | | | | | |
| | | | iNaturalist | | SUN | | Places | | Textures | | Average | |
| | | | AUROC↑ | FPR95↓ | AUROC↑ | FPR95↓ | AUROC↑ | FPR95↓ | AUROC↑ | FPR95↓ | AUROC↑ | FPR95↓ |
| 10,000 | ✓ | ✓ | 99.49 | 1.91 | 95.49 | 20.53 | 91.64 | 35.59 | 90.22 | 43.56 | 94.21 | 25.40 |
| 10,000 | ✗ | ✗ | 97.96 | 9.24 | 93.89 | 28.94 | 89.53 | 45.08 | 86.40 | 56.03 | 91.95 | 34.82 |
| 10,000 | ✗ | ✓ | 97.96 | 9.11 | 93.93 | 28.67 | 89.54 | 45.10 | 86.62 | 55.87 | 92.01 | 34.69 |
| 10,000 | ✓ | ✗ | 99.30 | 2.65 | 95.06 | 23.11 | 90.90 | 40.35 | 89.76 | 46.63 | 93.76 | 28.19 |
| 1,000 | ✓ | ✓ | 99.62 | 1.35 | 92.42 | 36.96 | 88.24 | 49.44 | 86.11 | 56.79 | 91.60 | 36.14 |
| 20,000 | ✓ | ✓ | 99.18 | 3.16 | 95.24 | 21.47 | 91.16 | 37.23 | 89.83 | 44.40 | 93.85 | 26.56 |
| 50,000 | ✓ | ✓ | 98.39 | 6.89 | 94.48 | 25.15 | 90.20 | 41.62 | 87.71 | 53.01 | 92.70 | 31.67 |

### A.4.1 IMPACT OF THE NUMBER OF SELECTED NEGATIVE LABELS.

We explore the impact of the number of selected negative labels on OOD detection. The results shown in Table 8 demonstrate that increasing the number of negative labels gradually improves the performance of OOD detection and then declines. This indicates that NegMining prioritizes selecting negative labels that are farther away from the ID labels, as they are more discriminative for detecting OOD images. This is consistent with the analysis in Section 3.3.

Table 9 shows the results of randomly selecting words in WordNet as negative labels. The comparison of Table 8 and Table 9 illustrates the role of NegMining algorithm. Concretely, when the number of negative label is less than 1,000, we notice that the OOD detection performances of random selection are better than NegMining. This is because that the negative labels selected by NegMining with higher rank (i.e., the top-$M$ ranked candidate labels when $M$ is small) are actually both far from ID and OOD samples, resulting in less divergences in $p_1$ and $p_2$ analysed in Section 3.3. Although the divergences are small, they can still provide a positive return in detecting OOD samples (i.e., AUROC $> 50\%$ for a binary classifier).

Furthermore, it is unfair to compare the performance of two methods only in the small $M$ case. With $M$ increases, the performances of random selection gradually converge to 92.18% AUROC as is performance bottleneck, while NegMining reaches a higher performance of 94.21% AUROC. The reason is that random selection indiscriminately introduces labels in WordNet, which may be toxic for OOD detection. In contrast, NegMining carefully selects words that are farther away from the ID semantics as negative labels, with a longer-term perspective. Therefore, the comparison of Table 8 and Table 9 reveals the effectiveness of the NegMining algorithm.

### A.4.2 DISCUSSION OF PERCENTILE DISTANCES.

To further investigate the impact of percentile distances, we conduct an ablation experiment, and the results are presented in Table 10. The study findings demonstrate that OOD detection performance improves significantly as the percentile decreases initially, but the rate of improvement gradually slows down, reaching its peak at the 5th percentile.

This phenomenon can be attributed to two main reasons. Firstly, ID labels that are closer to the candidate negative label are more likely to represent similar semantic meaning. As a result, by

Table 8: Impact of the number of negative labels $M$ selected by NegMining on ImageNet-1k benchmark. The negative labels are selected from WordNet. All values are percentages. ↑ indicates larger values are better and ↓ indicates smaller values are better.

| Negative Label Number | OOD Dataset | | | | | | | | Average | |
| | iNaturalist | | SUN | | Places | | Textures | | | |
| | AUROC↑ | FPR95↓ | AUROC↑ | FPR95↓ | AUROC↑ | FPR95↓ | AUROC↑ | FPR95↓ | AUROC↑ | FPR95↓ |
|---|---|---|---|---|---|---|---|---|---|---|
| 100 | 53.18 | 99.30 | 53.34 | 99.44 | 53.09 | 99.05 | 53.04 | 99.18 | 53.16 | 99.24 |
| 200 | 99.23 | 2.59 | 89.83 | 50.87 | 85.15 | 62.33 | 84.23 | 69.01 | 89.61 | 46.20 |
| 1,000 | 99.62 | 1.35 | 92.42 | 36.96 | 88.24 | 49.44 | 86.11 | 56.79 | 91.60 | 36.14 |
| 2,000 | 99.67 | 1.17 | 93.60 | 29.58 | 89.90 | 42.83 | 88.84 | 48.10 | 93.01 | 30.42 |
| 5,000 | 99.64 | 1.39 | 94.83 | 23.28 | 91.28 | 37.58 | 89.97 | 44.26 | 93.93 | 26.63 |
| 10,000 | 99.49 | 1.91 | 95.49 | 20.53 | 91.64 | 35.59 | 90.22 | 43.56 | 94.21 | 25.40 |
| 20,000 | 99.18 | 3.16 | 95.24 | 21.47 | 91.16 | 37.23 | 89.83 | 44.40 | 93.85 | 26.56 |
| 30,000 | 98.90 | 4.42 | 94.98 | 22.40 | 90.75 | 39.11 | 89.12 | 47.00 | 93.44 | 28.23 |
| 50,000 | 98.39 | 6.89 | 94.48 | 25.15 | 90.20 | 41.62 | 87.71 | 53.01 | 92.70 | 31.67 |

Table 9: Results of randomly selecting negative labels on ImageNet-1k benchmark. The negative labels are selected from WordNet. Each row of different experimental setting reports the average performance of 10 repeated runs, and the superscript represents its standard deviation.

| Random Selection | OOD Dataset | | | | | | | | Average | |
| | iNaturalist | | SUN | | Places | | Textures | | | |
| | AUROC↑ | FPR95↓ | AUROC↑ | FPR95↓ | AUROC↑ | FPR95↓ | AUROC↑ | FPR95↓ | AUROC↑ | FPR95↓ |
|---|---|---|---|---|---|---|---|---|---|---|
| 100 | $96.06^{\pm0.69}$ | $20.08^{\pm3.80}$ | $92.10^{\pm0.39}$ | $42.79^{\pm2.75}$ | $88.06^{\pm0.45}$ | $55.40^{\pm2.48}$ | $84.84^{\pm1.05}$ | $61.73^{\pm3.22}$ | $90.27^{\pm0.41}$ | $45.00^{\pm2.08}$ |
| 200 | $96.70^{\pm0.48}$ | $16.12^{\pm2.57}$ | $92.91^{\pm0.38}$ | $36.09^{\pm2.07}$ | $88.72^{\pm0.38}$ | $50.71^{\pm1.77}$ | $84.88^{\pm0.66}$ | $60.97^{\pm2.01}$ | $90.80^{\pm0.18}$ | $40.97^{\pm0.93}$ |
| 1,000 | $97.63^{\pm0.11}$ | $10.68^{\pm0.16}$ | $93.75^{\pm0.16}$ | $29.55^{\pm0.85}$ | $89.45^{\pm0.19}$ | $45.00^{\pm1.08}$ | $85.86^{\pm0.36}$ | $57.55^{\pm1.17}$ | $91.67^{\pm0.14}$ | $35.69^{\pm0.68}$ |
| 2,000 | $97.63^{\pm0.09}$ | $10.27^{\pm0.44}$ | $93.99^{\pm0.12}$ | $28.23^{\pm0.76}$ | $89.69^{\pm0.11}$ | $43.82^{\pm0.81}$ | $85.99^{\pm0.33}$ | $57.64^{\pm0.76}$ | $91.84^{\pm0.10}$ | $34.99^{\pm0.47}$ |
| 5,000 | $97.71^{\pm0.07}$ | $10.36^{\pm0.37}$ | $94.01^{\pm0.09}$ | $28.21^{\pm0.51}$ | $89.85^{\pm0.09}$ | $43.30^{\pm0.59}$ | $86.48^{\pm0.11}$ | $56.80^{\pm0.36}$ | $92.01^{\pm0.06}$ | $34.67^{\pm0.35}$ |
| 10,000 | $97.96^{\pm0.04}$ | $9.11^{\pm0.18}$ | $93.93^{\pm0.10}$ | $28.67^{\pm0.57}$ | $89.54^{\pm0.05}$ | $45.10^{\pm0.33}$ | $86.62^{\pm0.22}$ | $55.87^{\pm0.82}$ | $92.01^{\pm0.06}$ | $34.69^{\pm0.25}$ |
| 20,000 | $97.84^{\pm0.02}$ | $9.78^{\pm0.13}$ | $94.08^{\pm0.04}$ | $27.66^{\pm0.25}$ | $89.92^{\pm0.04}$ | $42.83^{\pm0.23}$ | $86.53^{\pm0.06}$ | $56.63^{\pm0.17}$ | $92.09^{\pm0.03}$ | $34.23^{\pm0.15}$ |
| 30,000 | $97.87^{\pm0.01}$ | $9.65^{\pm0.09}$ | $94.10^{\pm0.02}$ | $27.64^{\pm0.18}$ | $89.92^{\pm0.01}$ | $42.88^{\pm0.15}$ | $86.62^{\pm0.06}$ | $56.54^{\pm0.23}$ | $92.12^{\pm0.02}$ | $34.18^{\pm0.12}$ |
| 50,000 | $97.92^{\pm0.01}$ | $9.39^{\pm0.07}$ | $94.12^{\pm0.01}$ | $27.43^{\pm0.11}$ | $89.94^{\pm0.01}$ | $42.86^{\pm0.07}$ | $86.66^{\pm0.02}$ | $56.42^{\pm0.09}$ | $92.16^{\pm0.01}$ | $34.03^{\pm0.05}$ |

selecting the ID label as the anchor for the candidate negative label that is in close proximity to it, a more accurate measurement of the candidate negative label's distance from the entire ID label space can be obtained. Secondly, as illustrated in Fig. 2, there are several outliers in the ID label space. If these outliers are used as anchors, the distance between the candidate negative label and the entire ID label space would be inaccurate. To improve the quality of negative labels and enhance OOD detection performance, we use the ID label at the 5th percentile as the anchor. This approach strikes a balance between accurately measuring the distance between the candidate negative label and the ID label space while also avoiding outliers that could lead to inaccurate distance measurements.

### A.4.3 ABLATION ON CORPUS SOURCES OF NEGATIVE LABELS.

The role of the corpus is to provide a larger and more comprehensive semantic space, so that ID and OOD data can be characterized and distinguished in a more fine-grained manner in this space. We conduct ablative analysis using corpus sources for selecting negative labels and the results are shown in Table 11. We take Wikipedia dataset [4] containing cleaned articles of all languages. Part-of-Speech Tags dataset [5] is a 370k English words corpus.

In detail, we use the title field in Wikipedia dataset, which contains concepts or entities in the world. We first randomly sample 70,000 items as candidates, and then pick top-10,000 items as negative labels by applying NegMining algorithm. As for Part-of-Speech Tags dataset, it is processed like WordNet, i.e., only using nouns and adjectives in the dataset. We first randomly subsample the words to 70,000 as candidates, and then select top-10,000 words as negative labels through NegMining algorithm.

The results of Table 11 show that the proposed negative label is a robust scheme for zero-shot OOD detection, which can get reasonable performances on multiple corpora, and it is even robust across multiple languages.

---

[4] https://dumps.wikimedia.org/
[5] https://www.kaggle.com/datasets/ruchi798/part-of-speech-tagging

Table 10: The choice of distances for the selection of negative labels.

| Percentile | iNaturalist | | SUN | | Places | | Textures | | Average | |
|---|---|---|---|---|---|---|---|---|---|---|
| | AUROC↑ | FPR95↓ | AUROC↑ | FPR95↓ | AUROC↑ | FPR95↓ | AUROC↑ | FPR95↓ | AUROC↑ | FPR95↓ |
| 0.95 | 98.81 | 4.88 | 94.69 | 24.39 | 91.18 | 36.25 | 89.32 | 46.81 | 93.50 | 28.08 |
| 0.90 | 99.06 | 3.65 | 94.62 | 24.89 | 90.97 | 37.31 | 89.74 | 44.40 | 93.60 | 27.56 |
| 0.85 | 99.26 | 2.88 | 94.82 | 23.71 | 90.99 | 37.09 | 89.96 | 43.62 | 93.76 | 26.82 |
| 0.80 | 99.37 | 2.43 | 94.95 | 22.91 | 91.14 | 36.80 | 89.97 | 43.76 | 93.86 | 26.47 |
| 0.75 | 99.43 | 2.26 | 95.20 | 21.70 | 91.28 | 36.55 | 90.14 | 43.65 | 94.01 | 26.04 |
| 0.70 | 99.45 | 2.14 | 95.31 | 21.01 | 91.41 | 36.10 | 90.21 | 43.49 | 94.09 | 25.69 |
| 0.65 | 99.46 | 2.08 | 95.34 | 21.07 | 91.47 | 36.13 | 90.16 | 43.85 | 94.11 | 25.78 |
| 0.60 | 99.47 | 2.07 | 95.39 | 21.00 | 91.47 | 36.31 | 90.16 | 43.90 | 94.12 | 25.82 |
| 0.55 | 99.47 | 1.94 | 95.43 | 20.81 | 91.53 | 35.81 | 90.13 | 43.83 | 94.14 | 25.60 |
| 0.50 | 99.47 | 1.95 | 95.44 | 20.73 | 91.54 | 35.94 | 90.14 | 43.90 | 94.15 | 25.63 |
| 0.45 | 99.47 | 1.95 | 95.48 | 20.50 | 91.57 | 35.83 | 90.18 | 43.67 | 94.18 | 25.49 |
| 0.40 | 99.48 | 1.97 | 95.48 | 20.56 | 91.56 | 35.90 | 90.21 | 43.76 | 94.18 | 25.55 |
| 0.35 | 99.48 | 1.95 | 95.48 | 20.60 | 91.59 | 35.81 | 90.25 | 43.58 | 94.20 | 25.49 |
| 0.30 | 99.48 | 1.97 | 95.49 | 20.71 | 91.59 | 35.94 | 90.21 | 43.48 | 94.19 | 25.52 |
| 0.25 | 99.48 | 1.94 | 95.49 | 20.52 | 91.59 | 35.90 | 90.19 | 43.72 | 94.19 | 25.52 |
| 0.20 | 99.48 | 1.96 | 95.49 | 20.47 | 91.60 | 35.73 | 90.16 | 43.83 | 94.18 | 25.50 |
| 0.15 | 99.48 | 1.90 | 95.49 | 20.42 | 91.60 | 35.72 | 90.18 | 43.78 | 94.19 | 25.45 |
| 0.10 | 99.49 | 1.90 | 95.50 | 20.44 | 91.62 | 35.92 | 90.14 | 44.06 | 94.19 | 25.58 |
| 0.05 | 99.49 | 1.91 | 95.49 | 20.53 | 91.64 | 35.59 | 90.22 | 43.56 | 94.21 | 25.40 |
| 0.04 | 99.49 | 1.91 | 95.47 | 20.53 | 91.65 | 35.84 | 90.21 | 44.06 | 94.20 | 25.59 |
| 0.03 | 99.49 | 1.92 | 95.42 | 20.62 | 91.63 | 35.98 | 90.17 | 44.11 | 94.18 | 25.66 |
| 0.02 | 99.49 | 1.93 | 95.39 | 20.79 | 91.61 | 35.98 | 90.28 | 43.79 | 94.20 | 25.62 |
| 0.01 | 99.50 | 1.96 | 95.36 | 20.96 | 91.63 | 36.45 | 90.03 | 44.73 | 94.13 | 26.03 |
| 0.00 | 99.47 | 2.08 | 95.27 | 21.14 | 91.55 | 36.91 | 89.58 | 47.09 | 93.97 | 26.81 |

Table 11: Comparison with diffrent corpus sources.

| Source | Language | iNaturalist | | SUN | | Places | | Textures | | Average | |
|---|---|---|---|---|---|---|---|---|---|---|---|
| | | AUROC↑ | FPR95↓ | AUROC↑ | FPR95↓ | AUROC↑ | FPR95↓ | AUROC↑ | FPR95↓ | AUROC↑ | FPR95↓ |
| WordNet | English | 99.49 | 1.91 | 95.49 | 20.53 | 91.64 | 35.59 | 90.22 | 43.56 | 94.21 | 25.40 |
| Wikipedia | English | 96.07 | 20.89 | 93.08 | 36.05 | 90.09 | 47.48 | 73.38 | 86.68 | 88.16 | 47.78 |
| | French | 91.99 | 47.18 | 90.62 | 48.79 | 90.37 | 46.30 | 80.86 | 76.33 | 88.46 | 54.65 |
| Part-of-Speech Tags | English | 99.23 | 3.25 | 94.20 | 25.93 | 90.17 | 43.09 | 87.77 | 50.11 | 92.84 | 30.59 |

## A.5 Ablation study on the NegLabel score.

### A.5.1 The form of the NegLabel score.

In Section 3.2, we discuss about the scheme of designing zero-shot OOD score with negative labels:

$$S(\mathbf{x}) = S^*(\text{sim}(\mathbf{x}, \mathcal{Y}), \text{sim}(\mathbf{x}, \mathcal{Y}^-)). \tag{9}$$

The design criterion of $S^*$ can be summarized as follows: $S^*$ is positively correlated with the similarity of the sample in the ID label space and negatively correlated with the similarity of the sample in the negative label space.

Here, we provide two implementations, linear and proportional, as follows:

$$S_{\text{linear}}(\mathbf{x}) = T_1(\text{sim}(\mathbf{x}, \mathcal{Y})) - \alpha \cdot T_1(\text{sim}(\mathbf{x}, \mathcal{Y}^-)), \tag{10}$$

$$S_{\text{proportional}}(\mathbf{x}) = \frac{T_2(\text{sim}(\mathbf{x}, \mathcal{Y}))}{T_3(\text{sim}(\mathbf{x}, \mathcal{Y})) + T_3(\text{sim}(\mathbf{x}, \mathcal{Y}^-))}, \tag{11}$$

where $T_*(\cdot)$ are usually nonlinear transforms, which can map similarities to more discriminative values. And $\alpha$ is a factor that balances the number of ID and negative labels. Specially, when $T$ is an exponential-sum transform with temperature coefficient $\tau$, Eq. (11) becomes the sum-softmax form in Eq. (6).

Table 12 shows the results of different implementations for OOD score function. All experiments in this table are evaluated on ImageNet-1k benchmark without the grouping strategy. The results indicate that the proportional scores perform better than the linear scores. This is because the linear

form just simply fuses the similarities by linear combination, while the proportional form introduces an extra nonlinearity that normalize and stabilize the similarities.

Comparing row 5 and row 6, the result of sum-softmax score is better than max-softmax with negative labels. Because the sum-softmax measures the confidence that samples belong to the whole ID label space, which is more robust against ID inter-class confusion in zero-shot classification. Row 5 and row 7 show the comparison on whether to use exponential transform. It can be observed that using exponential function can significantly improve the performance. Because the image-text similarities in the CLIP-like model are smooth, as analyzed in Section 3.2.

In particular, the optimal temperature coefficient of sum-softmax score is $\tau = 0.01$, which makes the score act as a soft binarization function (each similarity is magnified 100 times and then scaled by exponential function, so that only the similarities near to the highest value are treated as positives). This is similar to our assumption in theoretical analysis (Section 3.3) that each negative label is a binary classifier. So, we provide some "binarized version" of the score functions, as shown in the Table 12 with $\mathbb{1}$ symbols. $\mathbb{1}_{[\cos(\cdot,\cdot)\geq\beta]}$ is a function to check if the cosine similarity is larger than $\beta$, and the value equals to 1 if the condition holds. $\beta$ is a binarization threshold and we set it to 0.25 for CLIP.

By replacing the exponential transform with the binarization transform (see row 5 and row 9), the results drop to 90.07% AUROC and 40.53% FPR95, but still show reasonable performances in zero-shot OOD detection. The binarized scores in linear form (refer to row 3 and row 4) also show the effectiveness of negative labels. Specially, the score function in row 4 is align with the "toy score function" $\widetilde{S}$ mentioned in theoretical analysis (Section 3.3), without directly using ID label to detect OOD samples. It still gets 78.12% AUROC, and illustrates the rationality of our theoretical analysis.

Table 12: The form of the NegLabel score. All experiments are evaluated on ImageNet-1k benchmark without the grouping strategy.

| No. | Mode | Score Function | iNaturalist | | SUN | | Places | | Textures | | Average | |
|---|---|---|---|---|---|---|---|---|---|---|---|---|
| | | | AUROC↑ | FPR95↓ | AUROC↑ | FPR95↓ | AUROC↑ | FPR95↓ | AUROC↑ | FPR95↓ | AUROC↑ | FPR95↓ |
| 1 | | $\sum_{i=1}^{K} \cos(\boldsymbol{h}, \boldsymbol{e}_i) - 0.1 \cdot \sum_{j=1}^{M} \cos(\boldsymbol{h}, \widetilde{\boldsymbol{e}}_j)$ | 98.14 | 9.33 | 80.26 | 80.63 | 68.68 | 89.29 | 78.02 | 82.80 | 81.27 | 65.51 |
| 2 | Linear | $\max_{i\in[1,2,\cdots,K]} \cos(\boldsymbol{h}, \boldsymbol{e}_i)$ | 87.75 | 66.27 | 89.10 | 54.81 | 88.35 | 51.90 | 82.30 | 72.02 | 86.87 | 61.25 |
| 3 | | $\sum_{i=1}^{K} \mathbb{1}_{[\cos(\boldsymbol{h},\boldsymbol{e}_i)\geq\beta]} - 0.1 \cdot \sum_{j=1}^{M} \mathbb{1}_{[\cos(\boldsymbol{h},\widetilde{\boldsymbol{e}}_j)\geq\beta]}$ | 95.64 | 9.34 | 86.12 | 44.74 | 83.93 | 49.59 | 75.62 | 62.06 | 85.33 | 41.43 |
| 4 | | $-\sum_{j=1}^{M} \mathbb{1}_{[\cos(\boldsymbol{h},\widetilde{\boldsymbol{e}}_j)\geq\beta]}$ | 98.97 | 3.47 | 75.98 | 78.36 | 62.04 | 89.44 | 75.47 | 72.62 | 78.12 | 60.97 |
| 5 | | $\dfrac{\sum_{i=1}^{K} e^{\frac{\cos(\boldsymbol{h},\boldsymbol{e}_i)}{\tau}}}{\sum_{i=1}^{K} e^{\frac{\cos(\boldsymbol{h},\boldsymbol{e}_i)}{\tau}}+\sum_{j=1}^{M} e^{\frac{\cos(\boldsymbol{h},\widetilde{\boldsymbol{e}}_j)}{\tau}}}$ | 99.30 | 2.65 | 95.06 | 23.11 | 90.90 | 40.35 | 89.76 | 46.63 | 93.76 | 28.19 |
| 6 | | $\dfrac{\max_{i\in[1,2,\cdots,K]} e^{\frac{\cos(\boldsymbol{h},\boldsymbol{e}_i)}{\tau}}}{\sum_{i=1}^{K} e^{\frac{\cos(\boldsymbol{h},\boldsymbol{e}_i)}{\tau}}+\sum_{j=1}^{M} e^{\frac{\cos(\boldsymbol{h},\widetilde{\boldsymbol{e}}_j)}{\tau}}}$ | 99.12 | 3.75 | 94.54 | 24.01 | 90.92 | 37.45 | 89.26 | 44.72 | 93.46 | 27.48 |
| 7 | Proportional | $\dfrac{\sum_{i=1}^{K} \cos(\boldsymbol{h},\boldsymbol{e}_i)}{\sum_{i=1}^{K} \cos(\boldsymbol{h},\boldsymbol{e}_i)+\sum_{j=1}^{M} \cos(\boldsymbol{h},\widetilde{\boldsymbol{e}}_j)}$ | 97.39 | 13.48 | 78.99 | 83.85 | 68.34 | 90.24 | 76.13 | 85.46 | 80.21 | 68.26 |
| 8 | | $\dfrac{\max_{i\in[1,2,\cdots,K]} \cos(\boldsymbol{h},\boldsymbol{e}_i)}{\sum_{i=1}^{K} \cos(\boldsymbol{h},\boldsymbol{e}_i)+\sum_{j=1}^{M} \cos(\boldsymbol{h},\widetilde{\boldsymbol{e}}_j)}$ | 97.92 | 10.38 | 94.03 | 28.24 | 89.05 | 44.34 | 90.94 | 40.51 | 92.98 | 30.87 |
| 9 | | $\dfrac{\sum_{i=1}^{K} \mathbb{1}_{[\cos(\boldsymbol{h},\boldsymbol{e}_i)\geq\beta]}}{\sum_{i=1}^{K} \mathbb{1}_{[\cos(\boldsymbol{h},\boldsymbol{e}_i)\geq\beta]}+\sum_{j=1}^{M} \mathbb{1}_{[\cos(\boldsymbol{h},\widetilde{\boldsymbol{e}}_j)\geq\beta]}}$ | 96.89 | 5.70 | 91.80 | 44.68 | 86.78 | 51.00 | 84.82 | 60.74 | 90.07 | 40.53 |

### A.5.2 COMPARISON WITH DIFFERENT TEMPERATURE VALUES OF SOFTMAX FUNCTION.

As analyzed in Section 3.2, the cosine function clamps the similarity score in $[-1, 1]$ and results in even smaller divergences between positive and negative pairs (e.g., the cosine similarities of positive pairs in CLIP are typically around 0.3 and that of negative pairs are around 0.1 (Ming et al., 2022b)). Such small differences may not adequately reflect the confidence variance of ID and OOD samples. Therefore, a suitable temperature coefficient $\tau$ is important to scale the similarity, allowing better reflecting the model's confidence.

We conduct ablation analysis towards the temperature coefficient $\tau$ in Eq. (6). The results in Table 13 illustrate that as the temperature coefficient $\tau$ increases, the performance of OOD detection shows an initial increase followed by a decrease trend, achieving the best performance at $\tau = 0.01$.

Table 13: Comparison with different temperature values of softmax function. The temperature values are sampled uniformly on the logarithmic axis.

| Temperature | OOD Dataset | | | | | | | | Average | |
|---|---|---|---|---|---|---|---|---|---|---|
| | iNaturalist | | SUN | | Places | | Textures | | | |
| | AUROC↑ | FPR95↓ | AUROC↑ | FPR95↓ | AUROC↑ | FPR95↓ | AUROC↑ | FPR95↓ | AUROC↑ | FPR95↓ |
| 0.0010 | 99.14 | 2.96 | 94.37 | 23.46 | 90.23 | 36.95 | 88.65 | 44.08 | 93.10 | 26.86 |
| 0.0013 | 99.19 | 2.91 | 94.53 | 23.44 | 90.55 | 36.96 | 89.01 | 44.24 | 93.32 | 26.89 |
| 0.0016 | 99.22 | 2.80 | 94.68 | 23.29 | 90.74 | 36.60 | 89.21 | 44.11 | 93.46 | 26.70 |
| 0.0020 | 99.27 | 2.70 | 94.82 | 22.80 | 90.94 | 36.23 | 89.39 | 43.67 | 93.61 | 26.35 |
| 0.0025 | 99.32 | 2.60 | 94.97 | 22.38 | 91.13 | 35.82 | 89.56 | 43.51 | 93.75 | 26.08 |
| 0.0032 | 99.36 | 2.39 | 95.12 | 21.73 | 91.33 | 35.32 | 89.74 | 42.89 | 93.89 | 25.58 |
| 0.0040 | 99.40 | 2.33 | 95.26 | 21.40 | 91.49 | 35.21 | 89.90 | 43.03 | 94.01 | 25.49 |
| 0.0050 | 99.44 | 2.22 | 95.37 | 20.81 | 91.61 | 35.36 | 90.04 | 42.82 | 94.11 | 25.30 |
| 0.0063 | 99.47 | 2.09 | 95.45 | 20.38 | 91.67 | 35.39 | 90.15 | 43.10 | 94.18 | 25.24 |
| 0.0079 | 99.49 | 1.90 | 95.49 | 20.58 | 91.64 | 35.82 | 90.23 | 43.72 | 94.21 | 25.51 |
| 0.0100 | 99.49 | 1.91 | 95.49 | 20.53 | 91.64 | 35.59 | 90.22 | 43.56 | 94.21 | 25.40 |
| 0.0126 | 99.52 | 1.86 | 95.44 | 21.18 | 91.35 | 38.19 | 90.23 | 45.28 | 94.14 | 26.63 |
| 0.0158 | 99.53 | 1.82 | 95.34 | 22.13 | 91.08 | 39.73 | 90.16 | 46.67 | 94.02 | 27.59 |
| 0.0200 | 99.46 | 2.06 | 93.13 | 37.88 | 87.12 | 58.11 | 87.88 | 59.34 | 91.90 | 39.35 |
| 0.0251 | 98.85 | 5.05 | 85.14 | 70.57 | 75.32 | 83.09 | 80.02 | 77.00 | 84.83 | 58.93 |
| 0.0316 | 99.14 | 2.98 | 94.36 | 23.79 | 90.22 | 36.82 | 88.65 | 44.61 | 93.09 | 27.05 |
| 0.0398 | 99.15 | 2.97 | 94.42 | 23.68 | 90.34 | 36.85 | 88.75 | 44.54 | 93.17 | 27.01 |
| 0.0501 | 99.17 | 2.94 | 94.48 | 23.44 | 90.43 | 36.86 | 88.87 | 44.54 | 93.23 | 26.94 |
| 0.0631 | 99.19 | 2.87 | 94.53 | 23.41 | 90.54 | 36.65 | 89.01 | 44.17 | 93.32 | 26.77 |
| 0.0794 | 99.21 | 2.85 | 94.61 | 23.24 | 90.65 | 36.61 | 89.11 | 44.24 | 93.39 | 26.73 |
| 0.1000 | 99.23 | 2.79 | 94.70 | 23.16 | 90.78 | 36.49 | 89.23 | 44.15 | 93.49 | 26.65 |
| 0.1259 | 99.27 | 2.75 | 94.82 | 22.74 | 90.94 | 36.26 | 89.38 | 44.06 | 93.60 | 26.45 |
| 0.1585 | 99.32 | 2.58 | 94.98 | 22.43 | 91.14 | 35.76 | 89.57 | 43.56 | 93.75 | 26.08 |
| 0.1995 | 99.38 | 2.39 | 95.17 | 21.55 | 91.38 | 35.34 | 89.79 | 42.87 | 93.93 | 25.54 |
| 0.2512 | 99.44 | 2.22 | 95.37 | 20.74 | 91.61 | 35.20 | 90.04 | 42.66 | 94.11 | 25.20 |
| 0.3162 | 99.49 | 1.92 | 95.49 | 20.59 | 91.65 | 35.82 | 90.23 | 43.67 | 94.22 | 25.50 |
| 0.3981 | 99.52 | 1.85 | 95.39 | 21.81 | 91.20 | 38.97 | 90.20 | 46.08 | 94.08 | 27.18 |
| 0.5012 | 99.52 | 1.75 | 94.72 | 26.92 | 89.82 | 46.69 | 89.58 | 51.19 | 93.41 | 31.64 |
| 0.6310 | 99.46 | 2.07 | 93.15 | 37.73 | 87.16 | 57.93 | 87.90 | 59.34 | 91.92 | 39.27 |
| 0.7943 | 99.33 | 2.67 | 90.90 | 50.85 | 83.67 | 69.90 | 85.48 | 66.60 | 89.84 | 47.50 |
| 1.0000 | 99.16 | 3.44 | 88.61 | 60.33 | 80.22 | 76.39 | 83.11 | 71.24 | 87.78 | 52.85 |
| 1.2589 | 99.00 | 4.36 | 86.65 | 66.66 | 77.40 | 80.79 | 81.28 | 74.33 | 86.08 | 56.53 |
| 1.5849 | 98.84 | 5.05 | 85.13 | 70.61 | 75.30 | 83.10 | 80.01 | 77.04 | 84.82 | 58.95 |
| 1.9953 | 98.70 | 5.84 | 83.96 | 73.06 | 73.76 | 84.56 | 79.16 | 78.72 | 83.90 | 60.55 |
| 2.5119 | 98.58 | 6.73 | 83.06 | 74.74 | 72.62 | 85.67 | 78.56 | 79.82 | 83.21 | 61.74 |
| 3.1623 | 98.47 | 7.45 | 82.36 | 76.15 | 71.75 | 86.43 | 78.13 | 80.57 | 82.68 | 62.65 |
| 3.9811 | 98.37 | 8.02 | 81.81 | 77.47 | 71.07 | 87.09 | 77.81 | 81.31 | 82.27 | 63.47 |
| 5.0119 | 98.28 | 8.48 | 81.37 | 78.27 | 70.55 | 87.60 | 77.57 | 81.95 | 81.94 | 64.08 |
| 6.3096 | 98.21 | 8.75 | 81.01 | 78.90 | 70.13 | 88.00 | 77.39 | 82.34 | 81.69 | 64.50 |
| 7.9433 | 98.15 | 9.06 | 80.73 | 79.37 | 69.81 | 88.28 | 77.24 | 82.70 | 81.48 | 64.85 |
| 10.0000 | 98.10 | 9.34 | 80.51 | 79.80 | 69.55 | 88.52 | 77.12 | 82.84 | 81.32 | 65.12 |

### A.5.3 EFFECT OF PROMPT ENGINEERING.

We evaluate NegLabel with different prompt template on the ImageNet-1k benchmark, as shown in Table 14. The first six rows are selected from the 80 prompt templates proposed by CLIP (Radford et al., 2021). The last two rows are our designed prompt templates.

Comparing the first three rows with the sixth row, it is evident that the performance of OOD detection significantly decreases when the prompt templates contain negative terms such as "dark", "blurry", and "low resolution". This may occur because OOD images might exhibit high similarity to prompts with negative words and ID labels, leading to misclassification. Conversely, when the prompt templates include positive terms like "good", the OOD detection performance improves noticeably. This is because OOD images are more likely to have low similarity with prompts containing positive words and ID labels, thereby enhancing the discrimination between ID and OOD images.

Furthermore, it can be observed from the fourth and fifth rows that excessive neutral words in neutral prompt templates tend to result in higher similarity between the negative labels and ID labels, thereby affecting the quality of selected negative labels. Therefore, we design two prompt templates with fewer neutral words and more positive adjectives as shown in the last two rows. Experimental results demonstrate that using "The nice <label>." as the prompt template achieves the best performance.

Table 14: Prompt engineering.

| Prompt | OOD Dataset | | | | | | | | | |
| | iNaturalist | | SUN | | Places | | Textures | | Average | |
| | AUROC↑ | FPR95↓ | AUROC↑ | FPR95↓ | AUROC↑ | FPR95↓ | AUROC↑ | FPR95↓ | AUROC↑ | FPR95↓ |
|---|---|---|---|---|---|---|---|---|---|---|
| A dark photo of a <label>. | 97.03 | 14.16 | 91.50 | 50.16 | 86.07 | 64.75 | 71.42 | 86.91 | 86.50 | 54.00 |
| A blurry photo of a <label>. | 98.64 | 5.74 | 91.26 | 47.36 | 85.68 | 61.76 | 79.14 | 71.70 | 88.68 | 46.64 |
| A low resolution photo of a <label>. | 99.39 | 2.74 | 93.06 | 37.79 | 88.80 | 52.88 | 78.33 | 77.50 | 89.89 | 42.73 |
| A cropped photo of a <label>. | 98.99 | 4.15 | 91.41 | 48.72 | 88.31 | 55.32 | 74.41 | 81.45 | 88.28 | 47.41 |
| A photo of a <label>. | 99.59 | 1.74 | 94.83 | 26.35 | 90.17 | 46.92 | 80.79 | 72.11 | 91.34 | 36.78 |
| A good photo of a <label>. | 99.53 | 2.01 | 95.32 | 22.71 | 90.39 | 43.99 | 84.09 | 61.90 | 92.33 | 32.65 |
| <label>. | 99.52 | 1.91 | 95.77 | 19.32 | 92.43 | 32.79 | 86.89 | 59.34 | 93.65 | 28.34 |
| The nice <label>. | 99.49 | 1.91 | 95.49 | 20.53 | 91.64 | 35.59 | 90.22 | 43.56 | 94.21 | 25.40 |

### A.5.4 EXPLORING THE GROUP NUMBERS OF THE GROUPING STRATEGY.

We investigate the impact of the hyperparameter (group number) in the grouping strategy and the corresponding results are presented in Table 15. We use $M = 10,000$ negative labels for all the experiments in this table, and the group number indicates how many subgroups should the negative labels be divided equally. The first row represents the baseline without using the grouping strategy when the group number is equal to one. The experimental results demonstrate that using the grouping strategy substantially enhances the OOD detection performance across four OOD datasets. Furthermore, as the group number gradually increases, the OOD detection performance remains stable, oscillating around 94.21% AUROC and 25.40% FPR95. These findings suggest that our grouping strategy is resilient to the hyperparameter (group number). Hence, we select group number as 100 for our main experiments.

Table 15: Impact of the group numbers in the grouping strategy on OOD detection. The experiments are conducted on ImageNet-1k benchmark.

| Group Number | OOD Dataset | | | | | | | | | |
| | iNaturalist | | SUN | | Places | | Textures | | Average | |
| | AUROC↑ | FPR95↓ | AUROC↑ | FPR95↓ | AUROC↑ | FPR95↓ | AUROC↑ | FPR95↓ | AUROC↑ | FPR95↓ |
|---|---|---|---|---|---|---|---|---|---|---|
| 1 | 99.30 | 2.65 | 95.06 | 23.11 | 90.90 | 40.35 | 89.76 | 46.63 | 93.76 | 28.19 |
| 50 | 99.48 | 1.94 | 95.47 | 20.63 | 91.62 | 35.91 | 90.19 | 43.56 | 94.19 | 25.51 |
| 100 | 99.49 | 1.91 | 95.49 | 20.53 | 91.64 | 35.59 | 90.22 | 43.56 | 94.21 | 25.40 |
| 150 | 99.49 | 1.89 | 95.52 | 20.47 | 91.66 | 35.70 | 90.20 | 43.95 | 94.22 | 25.50 |
| 200 | 99.49 | 1.90 | 95.50 | 20.58 | 91.66 | 35.66 | 90.24 | 43.60 | 94.22 | 25.43 |
| 250 | 99.50 | 1.88 | 95.55 | 20.22 | 91.65 | 35.69 | 90.18 | 43.97 | 94.22 | 25.44 |
| 300 | 99.50 | 1.85 | 95.53 | 20.32 | 91.60 | 35.95 | 90.14 | 44.18 | 94.19 | 25.58 |
| 350 | 99.51 | 1.84 | 95.54 | 20.32 | 91.58 | 35.94 | 90.13 | 44.26 | 94.19 | 25.59 |
| 400 | 99.50 | 1.90 | 95.50 | 20.61 | 91.66 | 35.76 | 90.25 | 43.62 | 94.23 | 25.47 |
| 450 | 99.50 | 1.89 | 95.53 | 20.48 | 91.67 | 35.73 | 90.21 | 43.92 | 94.23 | 25.50 |
| 500 | 99.52 | 1.80 | 95.57 | 20.20 | 91.55 | 36.33 | 90.02 | 44.95 | 94.16 | 25.82 |
| 1000 | 99.52 | 1.81 | 95.57 | 20.19 | 91.55 | 36.32 | 90.01 | 44.95 | 94.16 | 25.82 |

### A.5.5 EXPLORING THE RELATIONSHIP BETWEEN THE NUMBER OF FUSED GROUPS AND OOD DETECTION PERFORMANCE IN THE GROUPING STRATEGY.

Another question about grouping strategy is that why the performances increases by calculating the average OOD score over multiple independent and identically distributed subgroups. To explore this further, we conduct a series of ablation experiments with fusing different number of groups. Based on the 100-subgroup setting in Table 15, we randomly pick 1, 2, 5, 10, 20, and 50 subgroups, with 100 negative labels in each subgroup. And then we compute the average NegLabel score over the picked subgroups, repeating each experiment ten times for more accurate performances. The results in Table 16 indicate that OOD detection performance improves gradually with an increase in the number of selected groups. Additionally, the standard deviation generated from multiple experiments decreases progressively. This trend shows that the performance for each subgroup may be affected by the randomness of the negative labels. The average strategy of grouping helps NegLabel to stabilize the OOD score and reduce the variances of zero-shot OOD detection performances.

Table 16: Impact of the number of fused group in the grouping strategy on OOD detection. We choose the group number as 100 so the last row represents the original grouping strategy (baseline).

| Number of Fused Groups | iNaturalist | | SUN | | Places | | Textures | | Average | |
|---|---|---|---|---|---|---|---|---|---|---|
| | AUROC↑ | FPR95↓ | AUROC↑ | FPR95↓ | AUROC↑ | FPR95↓ | AUROC↑ | FPR95↓ | AUROC↑ | FPR95↓ |
| 1 | $98.90^{\pm0.018}$ | $4.15^{\pm0.177}$ | $92.93^{\pm0.098}$ | $37.27^{\pm0.515}$ | $89.14^{\pm0.118}$ | $50.65^{\pm0.813}$ | $87.81^{\pm0.275}$ | $54.26^{\pm1.082}$ | $92.20^{\pm0.056}$ | $36.58^{\pm0.233}$ |
| 2 | $98.89^{\pm0.032}$ | $4.20^{\pm0.181}$ | $92.85^{\pm0.085}$ | $38.11^{\pm0.539}$ | $89.13^{\pm0.116}$ | $50.55^{\pm0.803}$ | $87.66^{\pm0.0.242}$ | $55.07^{\pm0.767}$ | $92.13^{\pm0.074}$ | $36.98^{\pm0.422}$ |
| 5 | $99.35^{\pm0.013}$ | $2.69^{\pm0.081}$ | $94.33^{\pm0.037}$ | $27.62^{\pm0.382}$ | $90.41^{\pm0.072}$ | $42.57^{\pm0.490}$ | $88.87^{\pm0.112}$ | $48.71^{\pm0.576}$ | $93.24^{\pm0.044}$ | $30.39^{\pm0.255}$ |
| 10 | $99.43^{\pm0.013}$ | $2.26^{\pm0.067}$ | $94.91^{\pm0.036}$ | $23.30^{\pm0.297}$ | $90.99^{\pm0.038}$ | $38.61^{\pm0.288}$ | $89.41^{\pm0.119}$ | $45.74^{\pm0.545}$ | $93.69^{\pm0.036}$ | $27.48^{\pm0.149}$ |
| 20 | $99.47^{\pm0.008}$ | $2.07^{\pm0.049}$ | $95.20^{\pm0.027}$ | $21.68^{\pm0.218}$ | $91.32^{\pm0.026}$ | $37.04^{\pm0.195}$ | $89.80^{\pm0.046}$ | $44.41^{\pm0.362}$ | $93.95^{\pm0.013}$ | $26.30^{\pm0.154}$ |
| 50 | $99.48^{\pm0.004}$ | $1.98^{\pm0.043}$ | $95.42^{\pm0.018}$ | $20.72^{\pm0.168}$ | $91.56^{\pm0.018}$ | $36.23^{\pm0.191}$ | $90.12^{\pm0.042}$ | $43.75^{\pm0.223}$ | $94.15^{\pm0.012}$ | $25.67^{\pm0.131}$ |
| 100 | **99.49** | **1.91** | **95.49** | **20.53** | **91.64** | **35.59** | **90.22** | **43.56** | **94.21** | **25.40** |

### A.5.6 EXPLORING SIMILARITY CHOICES IN EQ. (5).

We tried using L1 distance and KL divergence to measure the similarity between images and negative labels, and the results are shown below. As the CLIP-like VLM models are trained under cosine similarity supervision, the features are naturally measured in the cosine space.

Table 17: Choices for similarity measurement between images and negative labels.

| Choices | iNaturalist | | SUN | | Places | | Textures | | Average | |
|---|---|---|---|---|---|---|---|---|---|---|
| | AUROC↑ | FPR95↓ | AUROC↑ | FPR95↓ | AUROC↑ | FPR95↓ | AUROC↑ | FPR95↓ | AUROC↑ | FPR95↓ |
| Cosine (baseline) | **99.49** | **1.91** | **95.49** | **20.53** | **91.64** | **35.59** | **90.22** | **43.56** | **94.21** | **25.40** |
| KL Divergence | 98.19 | 9.19 | 78.75 | 85.47 | 68.61 | 90.47 | 73.84 | 90.07 | 79.85 | 68.80 |
| L1 Distance | 97.34 | 11.95 | 80.28 | 79.39 | 69.29 | 89.14 | 66.57 | 86.51 | 78.37 | 66.75 |

Observing the experimental results, it can be seen that when using L1 distance and KL divergence as metrics, there is a significant drop in performance on SUN, Places, and Textures datasets, while the impact on iNaturalist dataset is relatively small. This is because our method (based on cosine similarity) achieves a 99.49 % AUROC on iNaturalist, almost completely distinguishing between ID and OOD data. Therefore, even when using metrics such as L1 and KL divergence that are not suitable for cosine space, there is still a significant difference between ID samples and OOD samples from iNaturalist. For more discussions on iNaturalist, please refer to Appendix B.1.

### A.6 ABLATION ON VLM ARCHITECTURES.

The detailed results are shown in Table 18.

Table 18: Comparison with different VLM architectures on ImageNet-1K (ID). All values are percentages. ↑ indicates larger values are better and ↓ indicates smaller values are better.

| Architecture | Backbone | iNaturalist | | SUN | | Places | | Textures | | Average | |
|---|---|---|---|---|---|---|---|---|---|---|---|
| | | AUROC↑ | FPR95↓ | AUROC↑ | FPR95↓ | AUROC↑ | FPR95↓ | AUROC↑ | FPR95↓ | AUROC↑ | FPR95↓ |
| | ResNet50 | 99.24 | 2.88 | 94.54 | 26.51 | 89.72 | 42.60 | 88.40 | 50.80 | 92.97 | 30.70 |
| | ResNet50x16 | 99.48 | 2.00 | 94.18 | 29.11 | 88.85 | 48.14 | 91.23 | 38.74 | 93.43 | 29.50 |
| CLIP | ViT-B/32 | 99.11 | 3.73 | 95.27 | 22.48 | 91.72 | 34.94 | 88.57 | 50.51 | 93.67 | 27.92 |
| | ViT-B/16 | 99.49 | 1.91 | 95.49 | 20.53 | 91.64 | 35.59 | 90.22 | 43.56 | 94.21 | 25.40 |
| | ViT-L/14 | 99.53 | 1.77 | 95.63 | 22.33 | 93.01 | 32.22 | 89.71 | 42.92 | 94.47 | 24.81 |
| ALIGN | EfficientNet-B7 | 98.86 | 4.55 | 90.93 | 43.18 | 85.03 | 61.01 | 88.24 | 49.89 | 90.76 | 39.66 |
| AltCLIP | ViT-L/14 | 99.73 | 1.18 | 95.74 | 21.44 | 93.42 | 30.11 | 89.50 | 41.81 | 94.60 | 23.63 |
| GroupViT | GroupViT | 98.07 | 8.60 | 91.52 | 35.12 | 88.85 | 41.63 | 88.45 | 47.06 | 91.72 | 33.10 |

### A.7 DISCUSSION ABOUT FORT ET AL. (2021).

Fort et al. (2021) use the ground-truth OOD labels for OOD detection, which is inconsistent with the OOD detection setting in real scenarios. It may not be appropriate and fair to compare Fort et al. (2021) in Table 1. The performance comparison between our method and Fort et al. (2021) is show in Table 19.

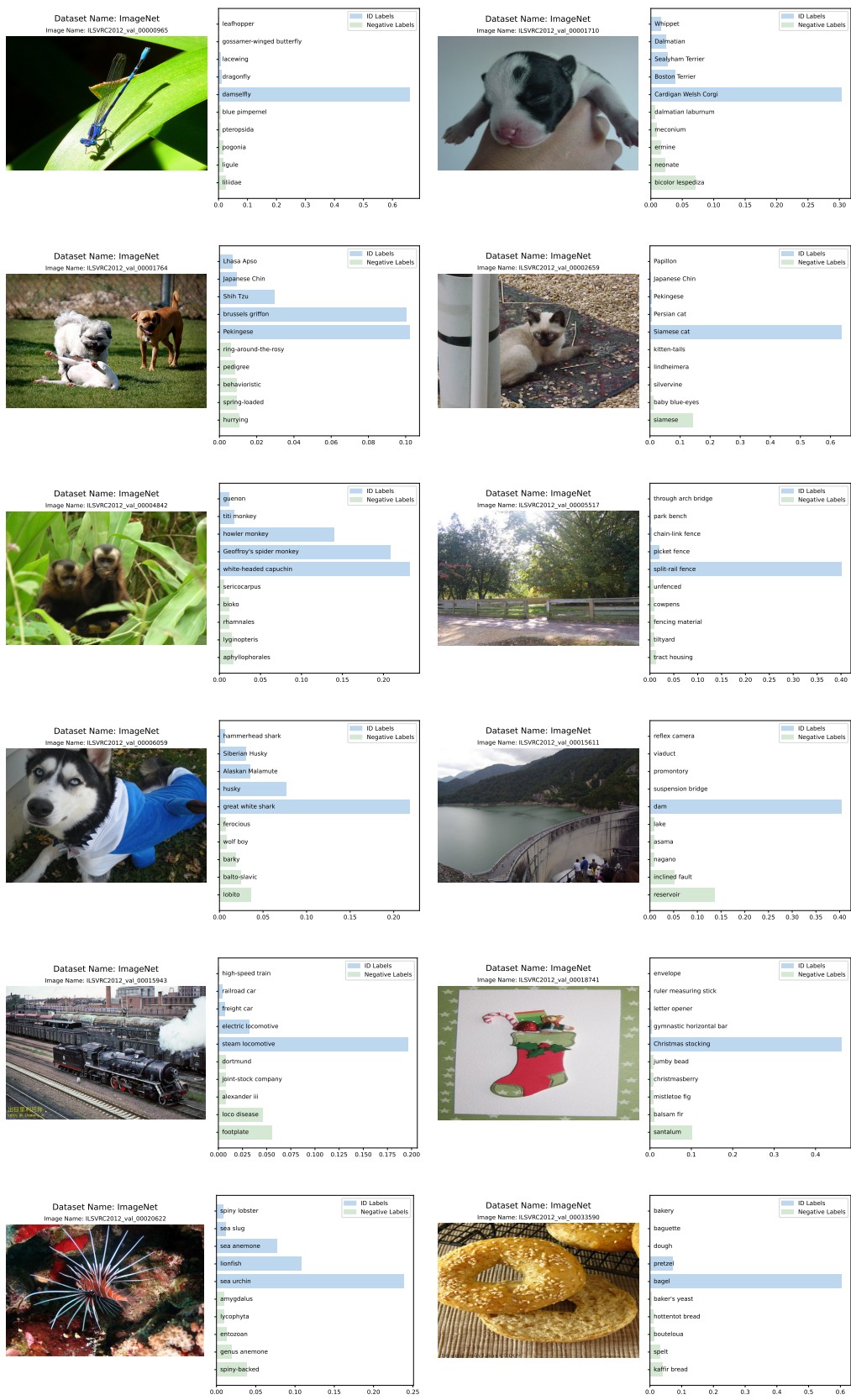

Figure 4: Case visualization of ID images. The left part of each subfigure contains the original image, its filename and its dataset name. The right part shows the softmax-normalized affinities among ID (blue) and negative (green) labels.

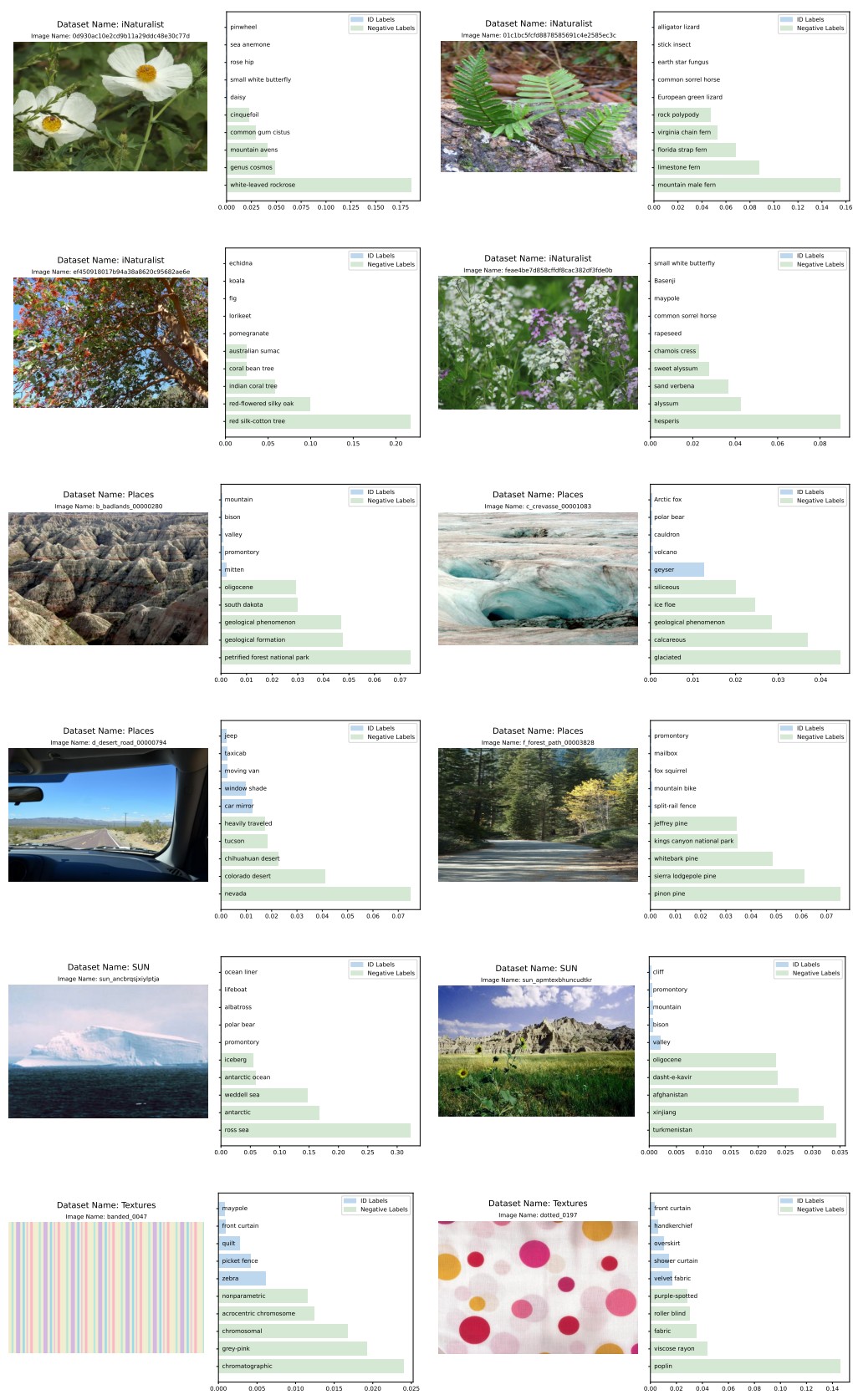

Figure 5: Case visualization of OOD images. The left part of each subfigure contains the original image, its filename and its dataset name. The right part shows the softmax-normalized affinities among ID (blue) and negative (green) labels.

In the original implementation of Fort et al. (2021), they directly apply softmax function on the cosine similarities between image and text embeddings, i.e., take temperature $\tau = 1$ in Eq. (6) in our paper. However, we observe that its performance on the ImageNet-1k benchmark is not as expected because the difference between positive matching and negative matching is too small, as mentioned in the discussion about the temperature coefficient of NegLabel Score in Section 3.2. We replace the temperature by $\tau = 0.01$ and achieved satisfactory results, with AUROC of 96.88% and FPR95 of 14.69%, exceeding all existing results (including zero-shot and non-zero-shot OOD detectors) reported on the ImageNet-1k benchmark. But we remind again that now the OOD detector works in an oracle situation because all OOD labels are added to the negative labels as known information.

Table 19: The performance comparison about Fort et al. (2021) on ImageNet-1k benchmark.

| Method | $\tau$ | iNaturalist | | SUN | | Places | | Textures | | Average | |
|---|---|---|---|---|---|---|---|---|---|---|---|
| | | AUROC | FPR95 | AUROC | FPR95 | AUROC | FPR95 | AUROC | FPR95 | AUROC | FPR95 |
| NegLabel | 0.01 | 99.49 | 1.91 | 95.49 | 20.53 | 91.64 | 35.59 | 90.22 | 43.56 | 94.21 | 25.40 |
| Fort et al. (2021) | 1 | 98.48 | 7.73 | 85.41 | 75.54 | 76.42 | 86.14 | 79.33 | 87.39 | 84.91 | 64.20 |
| Fort et al. (2021) | 0.01 | 99.57 | 1.47 | 98.37 | 7.11 | 94.28 | 25.32 | 95.30 | 24.88 | 96.88 | 14.69 |

## B    FURTHER ANALYSIS

### B.1    ANALYSIS OF THE PERFORMANCE ON INATURALIST

We think that the performance differences between iNaturalist and other OOD datasets mainly stem from the characteristics of the OOD datasets themselves. In Fig. 4 and Fig. 5, we provide a few sample images from the ID and OOD datasets for reference. The ID dataset ImageNet-1k contains a large number of categories related to animals and food, while iNaturalist consists of images of natural plants. SUN and Places datasets contain images of natural landscapes. Therefore, compared to other OOD datasets, iNaturalist has the largest differences in category terms compared to ImageNet, such as animals vs. plants. Therefore, introducing negative labels can significantly improve the OOD detection performance. On the other hand, SUN and Places datasets often contain multiple elements from the natural world beyond their annotated ground truth, making them more prone to confusion with the ID data.

### B.2    DISCUSSION ABOUT ZOC.

ZOC (Esmaeilpour et al., 2022) employs a captioner that has been pretrained on the COCO dataset (Lin et al., 2014) to generate multiple candidate labels for each image. These labels are then merged with the existing ID labels to calculate OOD scores. Consequently, the generated labels by ZOC are all derived from the COCO dataset, which poses a disadvantage when applying ZOC to large-scale datasets.

To address this limitation, we try to reproduce an upgraded version of ZOC on ImageNet-1k benchmark. We leveraged the capabilities of the multimodal model LLaVA (Liu et al., 2023) and ChatGPT to migrate ZOC to large-scale data sets. The idea comes from MCM's discussion and reproduction of ZOC.

We first used LLaVA to describe each sample and extract the entities contained in the description. Then we encountered a difficulty. LLaVA's descriptions of a large number of ID samples were slightly different from their corresponding ImageNet-1k labels, such as dog v.s. Alaskan Malamute. As a result, the ID label filtering policy cannot take effect. In order to overcome this problem, we collected all entities generated by LLaVA and used ChatGPT to filter words similar to the ImageNet-1k label. The remaining words are treated as candidate OOD labels of ZOC. The results shown in Table 20 demonstrate that our method NegLabel significantly outperforms ZOC.

There are several reasons for this phenomenon: Firstly, ZOC generates candidate out-of-distribution (OOD) labels for both in-distribution (ID) and OOD data. However, for ID data, the captioner may produce synonyms or closely related words to the ID labels, which can have high similarity with the image. This can lead to ID images being incorrectly classified as OOD. Secondly, the performance of ZOC heavily relies on the dataset used for pretraining the captioner. If the OOD data is not part of

Table 20: Performance comparison with ZOC.

| Method | iNaturalist | | SUN | | Places | | Textures | | Average | |
|---|---|---|---|---|---|---|---|---|---|---|
| | AUROC↑ | FPR95↓ | AUROC↑ | FPR95↓ | AUROC↑ | FPR95↓ | AUROC↑ | FPR95↓ | AUROC↑ | FPR95↓ |
| ZOC | 86.09 | 87.30 | 81.20 | 81.51 | 83.39 | 73.06 | 76.46 | 98.90 | 81.79 | 85.19 |
| NegLabel | 99.49 | 1.91 | 95.49 | 20.53 | 91.64 | 35.59 | 90.22 | 43.56 | 94.21 | 25.40 |

the captioner's pretrained categories, the generated candidate labels will be inaccurate, resulting in inaccurate scores for OOD data.

Compared to ZOC, our method, NegLabel, does not require additional complex pretraining processes and additional images. NegLabel is simpler to use and can easily generalize to different data and models. NegLabel also offers more stable and powerful performance of zero-shot OOD detection.

## B.3 DISCUSSION ABOUT CLIPN.

CLIPN utilizes additional datasets to train a text encoder that can understand negative prompts. It discriminates OOD data by comparing the similarity differences between the outputs of two text encoders and the image encoder. On the other hand, our method does not require additional data or training. It only introduces negative labels from the corpus to accomplish zero-shot OOD detection tasks. Therefore, in terms of the idea, both methods aim to introduce negative semantics of the ID category to reject inputs unrelated to the ID category. In terms of computational complexity, our method is slightly higher than CLIPN, but it is almost negligible compared to the computational complexity of CLIP itself. In terms of specific performance, our method has a higher average performance on the ImageNet-1k benchmark. Moreover, compared to CLIPN and NegLabel on CLIP-B/16 and CLIP-B/32, our method NegLabel is more robust to model architectures.

Table 21: Performance comparison with CLIPN.

| Method | iNaturalist | | SUN | | Places | | Textures | | Average | |
|---|---|---|---|---|---|---|---|---|---|---|
| | AUROC↑ | FPR95↓ | AUROC↑ | FPR95↓ | AUROC↑ | FPR95↓ | AUROC↑ | FPR95↓ | AUROC↑ | FPR95↓ |
| CLIPN (CLIP-B/32) | 94.67 | 28.75 | 92.85 | 31.87 | 86.93 | 50.17 | 87.68 | 49.49 | 90.53 | 40.07 |
| CLIPN (CLIP-B/16) | 95.27 | 23.94 | 93.93 | 26.17 | 90.93 | 40.83 | 92.28 | 33.45 | 93.10 | 31.10 |
| NegLabel (CLIP-B/32) | 99.11 | 3.73 | 95.27 | 22.48 | 91.72 | 34.94 | 88.57 | 50.51 | 93.67 | 27.92 |
| NegLabel (CLIP-B/16) | 99.49 | 1.91 | 95.49 | 20.53 | 91.64 | 35.59 | 90.22 | 43.56 | 94.21 | 25.40 |

## B.4 CASE ANALYSIS

To better illustrate the effectiveness of NegLabel, we provide some visualization results of ID and OOD images, as shown in Fig. 4 and Fig. 5. All the images are picked from the datasets in ImageNet-1k OOD detection benchmark. Each subfigure shows the original image and the top-5 affinities between the image and ID (blue) and negative (green) labels given by CLIP. The affinities are defined as softmax-normalized cosine similarities with temperature coefficient $\tau = 0.01$.

In Fig. 4, it is obvious that ID images have higher affinities on ID labels, and usually have low affinities on negative labels because of our proposed NegMining algorithm. For example, the first subfigure (row 1, column 1, ILSVRC2012_val_00000965) gets high affinity on its ground-truth ID label and gets low affinities on other labels, which is a good case of CLIP classifier, demonstrating a typical working scenario of NegLabel. However, CLIP does not always accurately determine which label an image belongs to. The image in row 4, column 1 (ILSVRC2012_val_00006059) shows a bad case of CLIP that the image is misclassified as "great white shark" and even confused between Alaskan and Husky. In fact, the image contains a husky wearing a shark coat, but suffers severe inter-class confusion, resulting in poor performances in MCM (Ming et al., 2022a), Energy (Liu et al., 2020), and etc. Fortunately, NegLabel with the sum-softmax score checks the affinities between the image and the whole ID label space, thus maintaining stable OOD detection performances even when the zero-shot classifier performs poorly. The same results can also be seen in row 2, column 1 (ILSVRC2012_val_00001764) and row 3, column 1 (ILSVRC2012_val_00004842).

In Fig. 5, OOD images usually have higher affinities on negative labels, because the negative labels contain various concepts and entities different from ID labels. As a zero-shot classifier, CLIP-like VLMs may not be good at fine-grained classification, making the affinities between OOD images and negative labels away from the one-hot distribution. As a result, the classification results for OOD images may be confused (like row 1, column 2, 01c1bc5fcfd8878585691c4e2585ec3c) or even incorrect (like row 2, column 1, ef450918017b94a38a8620c95682ae6e). But these negative labels tell us how "unlike" the image is from the ID data. Benefiting from the extensiveness of the large-scale corpus, almost all OOD images can have reasonable negative labels corresponding to them. This shows that NegLabel utilizes more knowledge from the text domain of VLMs and provides new clues for detecting OOD samples.

Some readers may be concerned about whether it is fair that the corpus contains a wide range of concepts, and may even directly cover the semantic labels of OOD samples. We think this is fair when the corpus is large enough, just as VLMs are reasonable for evaluation on zero-shot tasks, even though they may have seen this task-relevant data when they were trained. Besides, if developers have prior knowledge of potential OOD labels, they can manually include them in the negative label space to achieve better OOD detection performance.

## B.5 DETAILED DERIVATIONS

**About** $\mathrm{FPR}_\lambda$**.** We use $\mathrm{FPR}_\lambda$ as the performance metric of an OOD detector to show the separability between the ID and OOD samples, which is defined as

$$\mathrm{FPR}_\lambda = F_{\mathrm{out}}\left(F_{\mathrm{in}}^{-1}(\lambda)\right),\tag{12}$$

where $\lambda \in [0, 1]$ represents the true positive rate (TPR), which indicates the proportion of samples that are correctly classified as ID. The cumulative distribution functions (CDFs) $F_{\mathrm{in}}$ and $F_{\mathrm{out}}$ correspond to the scores obtained by the ID and OOD samples, respectively. The metric $\mathrm{FPR}_\lambda$ quantifies the degree of overlap between the scores assigned by the OOD detector to the ID and OOD samples, with lower values indicating better performance. Therefore, the $\mathrm{FPR}_\lambda$ metric, specifically in relation to the toy function $\widetilde{S}$, can be defined as follows:

$$
\begin{aligned}
\mathrm{FPR}_\lambda &= \Phi\left[\Phi^{-1}\left[\lambda; Mp_1, \sqrt{Mp_1(1-p_1)}\right]; Mp_2, \sqrt{Mp_2(1-p_2)}\right] \\
&= \frac{1}{2} + \frac{1}{2}\cdot\mathrm{erf}\left(\frac{\Phi^{-1}\left[\lambda; Mp_1, \sqrt{Mp_1(1-p_1)}\right] - Mp_2}{\sqrt{2Mp_2(1-p_2)}}\right) \\
&= \frac{1}{2} + \frac{1}{2}\cdot\mathrm{erf}\left(\sqrt{\frac{p_1(1-p_1)}{p_2(1-p_2)}}\,\mathrm{erf}^{-1}(2\lambda-1) + \frac{\sqrt{M}(p_1-p_2)}{\sqrt{2p_2(1-p_2)}}\right),
\end{aligned}
\tag{13}
$$

where

$$\Phi^{-1}\left[\lambda; Mp_1, \sqrt{Mp_1(1-p_1)}\right] = \sqrt{2Mp_1(1-p_1)}\,\mathrm{erf}^{-1}(2\lambda-1) + Mp_1.\tag{14}$$

**About the deviation of** $\mathrm{FPR}_\lambda$**.** We target on investigating the relationship between the $\mathrm{FPR}_\lambda$ metric and the number of negative labels $M$. Therefore, we calculate the partial derivative with respect to $M$ and use $z = \sqrt{\frac{p_1(1-p_1)}{p_2(1-p_2)}}\,\mathrm{erf}^{-1}(2\lambda-1) + \frac{\sqrt{M}(p_1-p_2)}{\sqrt{2p_2(1-p_2)}}$ to simplify the expression:

$$
\begin{aligned}
\frac{\partial \mathrm{FPR}_\lambda}{\partial M} &= \frac{1}{2}\cdot\frac{\partial\,\mathrm{erf}(z)}{\partial z}\cdot\frac{\partial z}{\partial M} \\
&= \frac{1}{2}\cdot\frac{2e^{-z^2}}{\sqrt{\pi}}\cdot\frac{p_1-p_2}{2\sqrt{2Mp_2(1-p_2)}} \\
&= \frac{e^{-z^2}}{2\sqrt{2\pi}}\cdot\frac{p_1-p_2}{\sqrt{Mp_2(1-p_2)}} < 0.
\end{aligned}
\tag{15}
$$

### B.6 MORE THEORETICAL ANALYSIS

In this section, we will consider a more general case for our analysis, where the probability that the label of $x$ is $\widetilde{y}_i$ is $p_i$, instead of $p$ used in Section 3.3. Thus, the positive count $c = \sum_i s_i^*$ will follow a Poisson binomial distribution instead of a binomial distribution. In the following, we find that Poisson binomial distribution can also be approximated by a Normal distribution based on the Lyapunov central limit theorem where the identical-distribution assumption made in the traditional central limit theorem is related.

Formally, let $S_i$ be a Bernoulli random variable with parameter $p_i$, which means that the probability of $S_i = 1$ is $p_i$ and the probability of $S_i = 0$ is $1 - p_i$. Here, we assume that, for any $i \in \mathbb{Z}^+$, $p_i \in [p_{\min}, p_{\max}]$, where $p_{\min} > 0$ and $p_{\max} < 1$.[6] Then, we let $C = S_1 + S_2 + \cdots + S_M$. Thus, we know $\mathbb{E}[S_i] = p_i$, $\text{Var}[S_i] = p_i(1 - p_i)$, $\mathbb{E}[C] = \sum_{i=1}^{M} p_i$, and $\text{Var}[C] = \sum_{i=1}^{M} p_i(1 - p_i)$. Based on the above expectation and variance values, we want to verify the Lyapunov condition: for some $\delta > 0$,

$$\lim_{M \to \infty} \frac{1}{(\text{Var}[C])^{3/2}} \sum_{i=1}^{M} \mathbb{E}\Big[|S_i - \mathbb{E}[S_i]|^{2+\delta}\Big] = 0. \tag{16}$$

We first analyze the term $\mathbb{E}[|S_i - \mathbb{E}[S_i]|^{2+\delta}]$ based on $S_i$ and $\mathbb{E}[S_i]$.

$$\begin{aligned}
\mathbb{E}\Big[|S_i - \mathbb{E}[S_i]|^{2+\delta}\Big] &= |1 - \mathbb{E}[S_i]|^{2+\delta}\text{Pr}(S_i = 1) + |0 - \mathbb{E}[S_i]|^{2+\delta}\text{Pr}(S_i = 0) \\
&= |1 - p_i|^{2+\delta}p_i + |0 - p_i|^{2+\delta}(1 - p_i) \\
&= (1 - p_i)^{2+\delta}p_i + p_i^{2+\delta}(1 - p_i) \\
&= (1 - p_i)p_i[(1 - p_i)^{1+\delta} + p_i^{1+\delta}] \tag{17}
\end{aligned}$$

Thus, we know $0 < \mathbb{E}[|S_i - \mathbb{E}[S_i]|^{2+\delta}] < 2$. Then, we analyze $(\text{Var}[C])^{3/2}$.

$$(\text{Var}[C])^{3/2} = \left(\sum_{i=1}^{M} p_i(1 - p_i)\right)^{3/2} \geq \left(\sum_{i=1}^{M} c_1\right)^{3/2} = c_1^{3/2}M^{3/2}, \tag{18}$$

where $c_1 = \min\{p_{\min} - p_{\min}^2, p_{\max} - p_{\max}^2\} > 0$. Based on Eq. (17) and Eq. (18), we have

$$0 < \frac{1}{(\text{Var}[C])^{3/2}} \sum_{i=1}^{M} \mathbb{E}\Big[|S_i - \mathbb{E}[S_i]|^{2+\delta}\Big] < \frac{2M}{c_1^{3/2}M^{3/2}} = \frac{2}{c_1^{3/2}\sqrt{M}}. \tag{19}$$

Thus, as $M \to \infty$,

$$0 < \frac{1}{(\text{Var}[C])^{3/2}} \sum_{i=1}^{M} \mathbb{E}\Big[|S_i - \mathbb{E}[S_i]|^{2+\delta}\Big] \to 0, \tag{20}$$

meaning that

$$\lim_{M \to \infty} \frac{1}{(\text{Var}[C])^{3/2}} \sum_{i=1}^{M} \mathbb{E}\Big[|S_i - \mathbb{E}[S_i]|^{2+\delta}\Big] = 0. \tag{21}$$

Hence, we verify the Lyapunov condition for $C$ and $S_i$. Based on the Lyapunov central limit theorem, we know that, as $M$ goes to infinity,

$$\frac{1}{\sqrt{\text{Var}[C]}} \sum_{i=1}^{M} \Big(S_i - \mathbb{E}[S_i]\Big) \xrightarrow{d} \mathcal{N}(0, 1), \tag{22}$$

where $\xrightarrow{d}$ means "converges in distribution". Thus, $C = \sum_{i=1}^{M} S_i$, a Poisson binomial random variable, approximately follows $\mathcal{N}(\sum_{i=1}^{M} \mathbb{E}[S_i], \text{Var}[C]) = \mathcal{N}(\mathbb{E}[C], \text{Var}[C])$, for a sufficiently large $M$. Then, following Section B.5, we can obtain the same result.

---

[6]It means that we only consider random variables with real randomness, i.e., these random variables have positive probabilities to at least take two values as their outputs. If $p_i = 0$ (or $p_i = 1$) for some $i$, then we know $S_i$ can only take the value of 0 (or 1) and has no probability to make $S_i = 1$ (or $S_i = 0$).

## C  ALGORITHMS

Here we give detailed implementations of the proposed algorithms. The central concept behind the NegMining algorithm (see Algorithm 2) is as follows: firstly, it calculates the $100\eta$-th percentile nearest distance between a negative candidate and the entire ID label space. Next, it selects the top-k candidates that have the greatest distances from the ID labels as the negative labels. This strategy guarantees that for every $y \in \mathcal{Y}$, there exists a substantial margin between the negative label and the ID label.

---

**Algorithm 2:** NegMining (Detailed Version)

**Input**  :Candidate labels $\mathcal{Y}^{\text{c}}$, ID labels $\mathcal{Y}$, Text encoder $\mathbf{f}^{\text{text}}$
**Output**:Negative labels $\mathcal{Y}^-$
// Calculate text embeddings
1 **for** $y_i \in \mathcal{Y}$ **do**
2 $\quad\big\lfloor\ e_i = \mathbf{f}^{\text{text}}(\text{prompt}(y_i));$
3 **for** $\widetilde{y}_i \in \mathcal{Y}^{\text{c}}$ **do**
4 $\quad\big|\quad \widetilde{e}_i = \mathbf{f}^{\text{text}}(\text{prompt}(\widetilde{y}_i));$
$\quad\big|\quad$ // Measure the distance from ID labels for each candidate label.
5 $\quad\big|\quad negsim_i = \{-\cos(\widetilde{e}_i, e_k)\}_{k=1}^{K};$
6 $\quad\big|\quad negsim_i' \leftarrow \text{reverse\_sort}(negsim_i);$
7 $\quad\big\lfloor\quad d_i \leftarrow$ the $100\eta$-th element in $negsim_i';$
// Pick $M$ negative labels with respect to the top-k distances.
8 $\left[d_1', d_2', \cdots, d_C'\right] = \text{reverse\_sort}([d_1, d_2, \cdots, d_C]);$
9 According to the rank of $\left[d_1', d_2', \cdots, d_C'\right]$, reorganize $\mathcal{Y}^{\text{c}}$ to $\mathcal{Y}^{'\text{c}}$;
10 $\mathcal{Y}^- \leftarrow$ first $M$ elements in $\mathcal{Y}^{'\text{c}}$.

---

The NegLabel score in a sum-softmax form is shown in Algorithm 3 and the NegLabel score with the grouping strategy is shown in Algorithm 4.

---

**Algorithm 3:** NegLabel Score

**Input**  : Model input $\mathbf{x}$, ID labels $\mathcal{Y}$, Negative labels $\mathcal{Y}^-$, Text encoder $\mathbf{f}^{\text{text}}$, Image encoder $\mathbf{f}^{\text{img}}$
**Output**: NegLabel score $S(\mathbf{x})$
// Preprocess:  calculate text embeddings.
1 **for** $y_i \in \mathcal{Y}$ **do**
2 $\quad\big\lfloor\ e_i = \mathbf{f}^{\text{text}}(\text{prompt}(y_i));$
3 **for** $\widetilde{y}_i \in \mathcal{Y}^-$ **do**
4 $\quad\big\lfloor\ \widetilde{e}_i = \mathbf{f}^{\text{text}}(\text{prompt}(\widetilde{y}_i));$
// Inference time.
5 $h = \mathbf{f}^{\text{img}}(\mathbf{x});$
6 $S(\mathbf{x}) = \dfrac{\sum\limits_{i=1}^{K} e^{\cos(h, e_i)/\tau}}{\sum\limits_{i=1}^{K} e^{\cos(h, e_i)/\tau} + \sum\limits_{j=1}^{M} e^{\cos(h, \widetilde{e}_j)/\tau}}.$

---

---

**Algorithm 4:** NegLabel Score with the Grouping Strategy

---

**Input** : Model input $\mathbf{x}$, ID labels $\mathcal{Y}$, Negative labels $\mathcal{Y}^-$, Text encoder $\mathbf{f}^{\text{text}}$, Image encoder $\mathbf{f}^{\text{img}}$, Number of grouping $n_g$

**Output :** NegLabel score $S(\mathbf{x})$

// Preprocess: calculate text embeddings.

1 **for** $y_i \in \mathcal{Y}$ **do**
2     $e_i = \mathbf{f}^{\text{text}}(\text{prompt}(y_i));$

3 **for** $\widetilde{y}_i \in \mathcal{Y}^-$ **do**
4     $\widetilde{e}_i = \mathbf{f}^{\text{text}}(\text{prompt}(\widetilde{y}_i));$

5 Divide the text embeddings of negative labels into $n_g$ groups.

// Inference time.

6 $h = \mathbf{f}^{\text{img}}(\mathbf{x});$

7 **for** *each group $l$* **do**

8     $s_l = \dfrac{\sum\limits_{i=1}^{K} e^{\cos(h, e_i)/\tau}}{\sum\limits_{i=1}^{K} e^{\cos(h, e_i)/\tau} + \sum\limits_{j=1}^{\lfloor M/n_g \rfloor} e^{\cos(h, \widetilde{e}_j)/\tau}}.$

9 $S(\mathbf{x}) = \frac{1}{n_g} \sum_{l=1}^{n_g} s_l$

---

