# OpenReview forum: "Negative Label Guided OOD Detection with Pretrained Vision-Language Models"
_ICLR.cc/2024/Conference — ICLR 2024 spotlight_

### Official Review · Reviewer_dUt1 · 2023-10-29

**Soundness:** 4 excellent
**Presentation:** 4 excellent
**Contribution:** 4 excellent
**Rating:** 8
**Confidence:** 4

**Summary:**

The field of out-of-distribution (OOD) detection seeks to recognize samples originating from unknown classes, thereby ensuring the trustworthiness of models when confronted with unanticipated inputs. While there exists a vast body of research delving into OOD detection within the visual modality, vision-language models (VLMs) stand out by synergizing both textual and visual data for a range of multi-modal tasks. However, there's a noticeable gap, with only a few OOD detection techniques capitalizing on the textual modality. In this study, the authors introduce a groundbreaking post hoc OOD detection technique, termed NegLabel, which harnesses a plethora of negative labels sourced from expansive corpus databases. They meticulously craft an innovative scheme, wherein the OOD score seamlessly collaborates with these negative labels. A detailed theoretical scrutiny aids in unraveling the intricate workings of these negative labels. Rigorous experimentation underscores that their NegLabel approach not only sets new standards in OOD detection benchmarks but also exhibits a laudable adaptability across diverse VLM architectures. Moreover, the NegLabel technique showcases exemplary resilience when faced with various domain shifts.

**Strengths:**

1 The research topic is very novel and important. There are a few papers focusing on this field. After reviewing this paper and relevant literature, it can be found that this paper moves a solid step to detect OOD samples when VLMs are available.

2 This paper identifies an important issue of relevant literature: how to select the negative words to improve the OOD detection power. Correspondingly, this paper proposes a novel method to address this important issue, which is a significant contribution in my view. The method's effectiveness is also verified by the extensive experiment.

3 The claims of this paper obtains strong support via extensive experiments that are based on various zero-shot OOD detection benchmarks and multiple VLM architectures.

4 The theoretical understanding starts from the data modelling, which is an elegant way to build up an understanding from the scratch. I am glad to see how the score is built up via a well-motivated data modelling method. The use of the probability tools is very interesting in the field, as we can directly see the distribution.

**Weaknesses:**

1 In Eq. (5), the cosine similarity is selected. Although the reason is given in the paper, it is still interesting to see if we can have other choices.

2 I enjoy reading how you move from basic data modelling to a more realistic data modelling in B.3. However, it would be better to point out why the contents in B.3 is more general, which can increase the interests of readers for understanding the proposed score.

3 In Table 2, all methods can obtain good results (AUROC>90% or FPR95<10% in the most cases). I actually do not see the necessarity to put that table in the main content. It would be better to move some interesting parts in Appendix to the main content.

4 What will happen if we select all different words from the ID classes?

**Questions:**

1 In Table 2, all methods can obtain good results (AUROC>90% or FPR95<10% in the most cases). I actually do not see the necessarity to put that table in the main content. Can we move it to appendix?

2 What will happen if we select all different words from the ID classes?

---

> ### Author Response · Authors · 2023-11-19
>
> >**W1:** In Eq. (5), the cosine similarity is selected. Although the reason is given in the paper, it is still interesting to see if we can have other choices.
>
> **A1:** Thanks for your suggestions! We tried using L1 distance and KL divergence to measure the similarity between images and negative labels, and the results are shown below. As the CLIP-like VLM models are trained under cosine similarity supervision, the features are naturally measured in the cosine space.
>
> Observing the experimental results, it can be seen that when using L1 distance and KL divergence as metrics, there is a significant drop in performance on SUN, Places, and Textures datasets, while the impact on iNaturalist dataset is relatively small. This is because our method (based on cosine similarity) achieves a 99.49 AUROC on iNaturalist, almost completely distinguishing between ID and OOD data. Therefore, even when using metrics such as L1 and KL divergence that are not suitable for cosine space, there is still a significant difference between ID samples and OOD samples from iNaturalist. For more discussions on iNaturalist, please refer to our response to reviewer 9gus in Reply **A1**.
>
> |       Choice      | iNaturalist |          |    SUN    |           |   Places  |           |  Textures |           |  Average  |           |
> |:-----------------:|:-----------:|:--------:|:---------:|:---------:|:---------:|:---------:|:---------:|:---------:|:---------:|:---------:|
> |                   |    AUROC↑   |  FPR95↓  |   AUROC↑  |   FPR95↓  |   AUROC↑  |   FPR95↓  |   AUROC↑  |   FPR95↓  |   AUROC↑  |   FPR95↓  |
> | Cosine (baseline) |  **99.49**  | **1.91** | **95.49** | **20.53** | **91.64** | **35.59** | **90.22** | **43.56** | **94.21** | **25.40** |
> |   KL Divergence   |    98.19    |   9.19   |   78.75   |   85.47   |   68.61   |   90.47   |   73.84   |   90.07   |   79.85   |   68.80   |
> |    L1 Distance    |    97.34    |   11.95  |   80.28   |   79.39   |   69.29   |   89.14   |   66.57   |   86.51   |   78.37   |   66.75   |
>
> >**W2:** I enjoy reading how you move from basic data modelling to a more realistic data modelling in B.3. However, it would be better to point out why the contents in B.3 is more general, which can increase the interests of readers for understanding the proposed score.
>
> **A2:** Thanks for your comments! In the theoretical analysis of the main text, to simplify the theoretical model and improve the readability of the article, we assume that the probability of a sample $x$ matching with each negative label is $p$. Therefore, for $M$ negative labels, we consider the number of matches between an image $x$ and them to follow a binomial distribution $B(M, p)$. However, due to the large number of negative labels, covering a wide semantic space, their affinity with the sample $x$ is variable.
>
> Specifically, in Appendix B.3, we assume that the probability of an image $x$ matching with a negative label is $p_i$, where different negative labels have different probabilities. This better reflects the real-world scenario, and thus we consider the number of matches between an image $x$ and them to follow a Poisson binomial distribution.
>
> We will provide more details in the main text to help readers better understand the general case in B.3.
>
> >**W3:** In Table 2, all methods can obtain good results (AUROC>90% or FPR95<10% in the most cases). I actually do not see the necessarity to put that table in the main content. It would be better to move some interesting parts in Appendix to the main content.
>
>
> **A3:** Thanks for your suggestions! We will reorganize our paper in the updated version.

---

> > ### Author Response · Authors · 2023-11-19
> >
> > >**W4:** What will happen if we select all different words from the ID classes?
> >
> > **A4:** Thanks for your comments! Here are the results (last row) that we select all different words from the ID classes.
> > | Negative Label Number | iNaturalist |        |   SUN  |        | Places |        | Textures |        | Average |        |
> > |-----------------------|:-----------:|:------:|:------:|:------:|:------:|:------:|:--------:|:------:|:-------:|:------:|
> > |                       |    AUROC↑   | FPR95↓ | AUROC↑ | FPR95↓ | AUROC↑ | FPR95↓ |  AUROC↑  | FPR95↓ |  AUROC↑ | FPR95↓ |
> > | 100                   |    53.18    |  99.30 |  53.34 |  99.44 |  53.09 |  99.05 |   53.04  |  99.18 |  53.16  |  99.24 |
> > | 200                   |    99.23    |  2.59  |  89.83 |  50.87 |  85.15 |  62.33 |   84.23  |  69.01 |  89.61  |  46.20 |
> > | 1,000                 |    99.62    |  1.35  |  92.42 |  36.96 |  88.24 |  49.44 |   86.11  |  56.79 |  91.60  |  36.14 |
> > | 2,000                 |    99.67    |  1.17  |  93.60 |  29.58 |  89.90 |  42.83 |   88.84  |  48.10 |  93.01  |  30.42 |
> > | 5,000                 |    99.64    |  1.39  |  94.83 |  23.28 |  91.28 |  37.58 |   89.97  |  44.26 |  93.93  |  26.63 |
> > | 10,000                |    99.49    |  1.91  |  95.49 |  20.53 |  91.64 |  35.59 |   90.22  |  43.56 |  94.21  |  25.40 |
> > | 20,000                |    99.18    |  3.16  |  95.24 |  21.47 |  91.16 |  37.23 |   89.83  |  44.40 |  93.85  |  26.56 |
> > | 30,000                |    98.90    |  4.42  |  94.98 |  22.40 |  90.75 |  39.11 |   89.12  |  47.00 |  93.44  |  28.23 |
> > | 50,000                |    98.39    |  6.89  |  94.48 |  25.15 |  90.20 |  41.62 |   87.71  |  53.01 |  92.70  |  31.67 |
> > | All                   |    97.95    |  9.18  |  94.16 |  27.26 |  89.93 |  42.78 |   86.67  |  56.44 |  92.18  |  33.91 |

---

> > > ### Comment · Reviewer_dUt1 · 2023-11-22
> > >
> > > Thanks for the detailed response and clarification of the authors. My concerns have been addressed. Overall, I think it is a good paper and I will raise my score to support this paper.

---

> > > > ### Author Response · Authors · 2023-11-22
> > > > **Thanks for raising your score!**
> > > >
> > > > We sincerely appreciate the time and effort you invested in reviewing our manuscript. Your insightful comments and suggestions have been invaluable in enhancing the quality of our work. We are grateful for your constructive feedback, which has significantly contributed to the improvement of our paper. Thank you for your guidance and support.

---

### Official Review · Reviewer_9gus · 2023-10-30

**Soundness:** 4 excellent
**Presentation:** 4 excellent
**Contribution:** 4 excellent
**Rating:** 8
**Confidence:** 4

**Summary:**

In the traditional OOD detection, the text information of labels will be abandoned, which actually loses some information regarding labels. With the assist of VLMs, this kind of information might be useful for OOD detection. The authors present a study aligning with this interesting and new research direction: OOD detection with VLMs. The existing methods are quite on the early stage, and this paper challenges these methods and proposes a new OOD detection method consisting of a negative label selection method and a new score function. Experiments are solid and form a solid contribution to OOD detection with VLMs.

**Strengths:**

+ Experiments are solid and cover many aspects, which can address many in-the-mind issues
automatically. It is enjoyable to the whole experimental section, which also contains some interesting insights directly from the experiments conducted in this paper.

+ The idea to design the new score function is straightforward after demonstrating an intuitive motivation. The motivation of this paper is supported by some evidence instead of "wording", which appreciated.

+ The presentation is generally good. The flow is great to follow, and necessary analysis is easy to follow as well.

**Weaknesses:**

- OOD detection methods have different performance on different datasets. On some datasets, the performance is extremely good. Can we see the difference among these datasets? Why can OOD detection be easily addressed on some datasets?

- Especially for results based on CLIP-B/16 with various ID datasets, the performance is too high. More explanation is need.

- Figures might be better to illustrate the final performance in Section 4.

- What is the relationship between Bernoulli distribution and binomial distribution? This is a well-known result in statistics?

- Why is B.3 more general than things demonstrated in main context?

- Please keep the naming strategy consistent in Algorithm 1 and other algorithms.

**Questions:**

In general, this paper addresses a significant problem and makes a solid contribution to this field. However, please answer questions listed in the Weakness:

- OOD detection methods have different performance on different datasets. On some datasets, the performance is extremely good for all methods. Can we see the difference among these datasets? Why can OOD detection be easily addressed on some datasets?

- What is the relationship between Bernoulli distribution and binomial distribution? This is a well-known result in statistics?

- Why is B.3 more general than things demonstrated in main context?

---

> ### Author Response · Authors · 2023-11-19
>
> >**W1:** OOD detection methods have different performance on different datasets. On some datasets, the performance is extremely good. Can we see the difference among these datasets? Why can OOD detection be easily addressed on some datasets?
>
>
> **A1:** Thanks for your comments! We think that the performance differences mainly stem from the characteristics of the OOD datasets themselves. In  Figure 4 and Figure 5 of the Appendix, we provide a few sample images from the ID and OOD datasets for reference. The ID dataset ImageNet contains a large number of categories related to animals and food, while iNaturalist consists of images of natural plants. SUN and Places datasets contain images of natural landscapes. Therefore, compared to other OOD datasets, iNaturalist has the largest differences in category terms compared to ImageNet, such as animals vs. plants. Therefore, introducing negative labels can significantly improve the OOD detection performance. On the other hand, SUN and Places datasets often contain multiple elements from the natural world beyond their annotated ground truth, making them more prone to confusion with the ID data.
>
> Furthermore, recent research trends have also shown that achieving high performance on iNaturalist is quite common. For example, ASH [1] and NNGuide [2] both achieve over 97% AUROC on iNaturalist, while GEN [3] achieves 99.13% AUROC.
>
> >**W2:** Especially for results based on CLIP-B/16 with various ID datasets, the performance is too high. More explanation is need.
>
> **A2:** Thanks for your suggestions! We follow MCM to conduct experiments on the large-scale benchmark in Table 1 and small-scale benchmarks in Table 2. The performance of MCM in Table 2 is very high and our NegLabel outperforms MCM. Our opinion about the high performance is consistent to MCM, i.e., the small-scale ID datasets have less challenges for VLMs.
>
> >**W3:** Figures might be better to illustrate the final performance in Section 4.
>
> **A3:** Thanks for your suggestions! We will replace some tables with figures in Section 4 in the updated paper.
>
> >**W4:** What is the relationship between Bernoulli distribution and binomial distribution? This is a well-known result in statistics?
>
> **A4:** The Bernoulli distribution is a special case of the binomial distribution, where n = 1. Symbolically, X ~ B(1, p) has the same meaning as X ~ Bernoulli(p). Conversely, any binomial distribution, B(n, p), is the distribution of the sum of n independent Bernoulli trials, Bernoulli(p), each with the same probability p.[4]
>
>
>
> >**W5:** Why is B.3 more general than things demonstrated in main context?
>
> **A5:** Thanks for your comments! In the theoretical analysis of the main text, to simplify the theoretical model and improve the readability of the article, we assume that the probability of a sample $x$ matching with each negative label is $p$. Therefore, for $M$ negative labels, we consider the number of matches between an image $x$ and them to follow a binomial distribution $B(M, p)$. However, due to the large number of negative labels, covering a wide semantic space, their affinity with the sample $x$ is variable.
>
> Specifically, in Appendix B.3, we assume that the probability of an image $x$ matching with a negative label is $p_i$, where different negative labels have different probabilities. This better reflects the real-world scenario, and thus we consider the number of matches between an image $x$ and them to follow a Poisson binomial distribution.
>
> We will provide more details in the main text to help readers better understand the general case in B.3.
>
> >**W6:** Please keep the naming strategy consistent in Algorithm 1 and other algorithms.
>
> **A6:** Thanks for your suggestions! We will revise the algorithms to keep the naming strategy consistent in the updated paper.
>
> **Reference**
>
> [1] Djurisic, Andrija, et al. "Extremely Simple Activation Shaping for Out-of-Distribution Detection." The Eleventh International Conference on Learning Representations. 2022.
>
> [2] Park, Jaewoo, Yoon Gyo Jung, and Andrew Beng Jin Teoh. "Nearest Neighbor Guidance for Out-of-Distribution Detection." Proceedings of the IEEE/CVF International Conference on Computer Vision. 2023.
>
> [3] Liu, Xixi, Yaroslava Lochman, and Christopher Zach. "GEN: Pushing the Limits of Softmax-Based Out-of-Distribution Detection." Proceedings of the IEEE/CVF Conference on Computer Vision and Pattern Recognition. 2023.
>
> [4] https://en.wikipedia.org/wiki/Binomial_distribution#Bernoulli_distribution

---

> > ### Comment · Reviewer_9gus · 2023-11-22
> > **Thanks for the reponse.**
> >
> > Thanks for the detailed response from the authors. My main concerns have been addressed and will keep my initial score.

---

### Official Review · Reviewer_JPmP · 2023-10-30

**Soundness:** 3 good
**Presentation:** 3 good
**Contribution:** 3 good
**Rating:** 6
**Confidence:** 4

**Summary:**

This paper is a pioneer work to study the OOD detection problem under VLMs. Compared to state-of-the-art papers in this frontier, this paper addresses several issues that SOTA methods do not cover. These issues are justified well in this paper, and the corresponding explanation is reasonable and solid. Then, a OOD-word selection method and a corresponding score function is proposed to address the observed issues. The proposed method makes sense and works well in the extensive experiments.

**Strengths:**

1 As a pioneer study in this field, I find that this paper is easy to follow and demonstrates issues of SOTA methods very clearly. Thus, in my point of view, this paper has a great potential to motivate more relevant work in this important field: OOD detection with VLMs/Zero-shot OOD detection.

2 I am convinced that the proposed work has a solid contribution by the extensive experiments conducted by the authors. The experiments cover many scenarios, which is a solid evidence that the proposed method has its own contribution to this field.

3 This paper also benefits from the theoretical understanding of the proposed score function. The analysis is easy to follow and looks totally new to me.

**Weaknesses:**

1 Figure 1 is clear and important. However, it makes me think about one question: in a statistical view, what should the right subfigure be?

2 In Algorithm 1, the comments are too long. It would be better to say the key functionality of this part.

3 Why does grouping strategy appear in Section 3.2 instead of 3.1? It seems that Grouping Strategy is a more advanced way to pre-process negative labels?

4 Can the authors explain why is the case considered in B.3 more general? More explanation can make readers know your analysis better.

**Questions:**

1 Figure 1 is clear and important. However, it makes me think about one question: in a statistical view, what should the right subfigure be?

2 Why does grouping strategy appear in Section 3.2 instead of 3.1? It seems that Grouping Strategy is a more advanced way to pre-process negative labels?

3 Can the authors explain why is the case considered in B.3 more general? More explanation can make readers know your analysis better.

---

> ### Author Response · Authors · 2023-11-19
>
> >**W1:** Figure 1 is clear and important. However, it makes me think about one question: in a statistical view, what should the right subfigure be?
>
>
> **A1:** Thanks for the comments! The actual subgraph would be very difficult to visualize, as the x-axis would contain 1,000 (ID labels) + 10,000 (negative labels) elements. We have provided the visualization results of the top 5 ID labels and negative labels similarities in Figures 4 and 5 in the appendix.
>
>
> >**W2:** In Algorithm 1, the comments are too long. It would be better to say the key functionality of this part.
>
> **A2:** Thanks for your suggestions! We will revise the comments in Algorithm 1.
>
> >**W3:** Why does grouping strategy appear in Section 3.2 instead of 3.1? It seems that Grouping Strategy is a more advanced way to pre-process negative labels?
>
> **A3:** Thanks for this question. Here we detail our proposed pipeline of Negative-Label-based OOD detection: corpus -> NegMining -> Grouping -> NegLabel Score. Specifically, our method requires first selecting appropriate negative labels through NegMining (as shown in Section 3.1), and then designing a NegLabel score using these negative labels (as shown in Section 3.2 part 1). The Grouping strategy is an optimization of the NegLabel score. By grouping the selected negative labels and calculating the NegLabel score for each group, the average of all groups is taken as the final OOD score. Therefore, the Grouping Strategy is not a more advanced way to preprocess negative labels. It does not involve the preprocessing process of negative labels. It is suitable to be placed in Section 3.2 part 2 as it optimizes and enhances the OOD score.
>
> To better understand the grouping strategy, we provide a toy example. Let's assume for an ID image, there is a relatively high response in the ID labels, but coincidentally, there is also a relatively high response in the negative labels, with both responses being of comparable magnitude. Without grouping, after softmax function, the probabilities for ID and negative labels might both be around 0.5, resulting in a score of 0.5.
>
> However, if we divide the negative labels into 10 groups, this misclassification would be included in one of the groups at most. The score for this group would be 0.5, while the rest of the groups would be close to 1.0. Thus, the final score would be calculated as 1 * 0.9 + 0.5 * 0.1 = 0.95. This illustrates how the Grouping Strategy helps mitigate the impact of such misclassifications.
>
> >**W4:** Can the authors explain why is the case considered in B.3 more general? More explanation can make readers know your analysis better.
>
> **A4:** Thanks for your comments! In the theoretical analysis of the main text, to simplify the theoretical model and improve the readability of the article, we assume that the probability of a sample $x$ matching with each negative label is $p$. Therefore, for $M$ negative labels, we consider the number of matches between an image $x$ and them to follow a binomial distribution $B(M, p)$. However, due to the large number of negative labels, covering a wide semantic space, their affinity with the sample $x$ is variable.
>
> Specifically, in Appendix B.3, we assume that the probability of an image $x$ matching with a negative label is $p_i$, where different negative labels have different probabilities. This better reflects the real-world scenario, and thus we consider the number of matches between an image $x$ and them to follow a Poisson binomial distribution.
>
> We will provide more details in the main text to help readers better understand the general case in B.3.

---

### Official Review · Reviewer_RPtB · 2023-11-03

**Soundness:** 4 excellent
**Presentation:** 4 excellent
**Contribution:** 3 good
**Rating:** 8
**Confidence:** 5

**Summary:**

This paper introduces a novel method for zero-shot out-of-distribution (OOD) detection, leveraging the capabilities of vision-language models (VLMs). The primary objective is to enhance OOD detection by incorporating textual information. NegLabel introduces a large number of negative labels, extending the label space to distinguish OOD samples more effectively. The method uses a novel scheme for the OOD score, taking into account affinities between images and both ID labels and negative labels. The paper provides a theoretical analysis, justifying the use of negative labels.

**Strengths:**

- **Well-written and technically sound paper** The paper is well-structured and its concepts and contributions are clearly presented.

- **Quality illustrations**

- **Extensive experiments** The paper conducts extensive experiments, including OOD detection, hard OOD detection, and robustness to domain shifts.

- **Large and consistent gains compared to several SOTA baselines** The paper's results show substantial improvements over several state-of-the-art (SOTA) baselines.

- **Intuitive justification and theoretical analysis provided** The paper offers an intuitive justification for the proposed method and includes theoretical analysis to explain the mechanism of negative labels.

- **Extensive ablations**

**Weaknesses:**

- **Assumption about CLIP's latent space** The assumption that ID image and corresponding ID label embeddings are close in CLIP's latent space is actually false. The contrastive learning strategy of CLIP is based on cosine similarity and thus textual and visual embeddings are actually on different manifolds see (Weixin Liang et al. 2022) for more details. Using ID labels as proxies for the ID images may thus produce unexpected results. In my understanding, the negative labels are compared to the images using cosine similarity, and thus having them close to ID textual embeddings still makes sense but the authors should be cautious here.

- **Comparison with related work** The method is close in spirit to CLIPN (Wang et al. 2023) as well as ZOC (Esmaeilpour et al 2022). The former fine-tunes CLIP to incorporate negative prompts to assess the probability of a concept not being present in the image. An extensive comparison with CLIPN would be overkill but some discussion about it would be welcome.  The second also performs zero-shot OOD using maximum softmax probability on the labels extended with the object detected in the images. Comparisons with ZOC are presented in the supplementary but would better belong in the main paper IMO as it is one of the few recent zero-shot OOD detection baselines. To be fair this comparison should be provided with the same set-up as in Table 2.

- **Misleading baseline presentation** Even if the MSP, ODIN, or Energy Logit baseline are generally used on models fine-tuned on a downstream task, one could also use these detectors in a zero-shot setting. Classifying them in purely zero-shot can be misleading eg. MCM can be seen as zero-shot MSP on CLIP's output probabilities.

- **Short Related Work** The related work section is relatively short, especially for such a prolific field as OOD detection. I don't think it is really detrimental to the paper but a discussion on the sense of OOD detection for foundation models trained on vast and various data would be welcome.

I am willing to improve my rating depending on the author's rebuttal.

**References**

Wang, H., Li, Y., Yao, H., & Li, X. (2023). CLIPN for Zero-Shot OOD Detection: Teaching CLIP to Say No, ICCV 2023.

Esmaeilpour, S., Liu, B., Robertson, E., & Shu, L. (2022). Zero-Shot Out-of-Distribution Detection Based on the Pre-trained Model CLIP. Proceedings of the AAAI Conference on Artificial Intelligence, 36(6), Article 6.

Liang, W., Zhang, Y., Kwon, Y., Yeung, S., & Zou, J. (2022, May 16). Mind the Gap: Understanding the Modality Gap in Multi-modal Contrastive Representation Learning. Advances in Neural Information Processing Systems.

**Questions:**

- I wonder why the authors did not add the simple and strong baseline Mahaloanobis (Lee et al. 2018) /SSD (Sehwag et al. 2022).
- No ablations are provided on $n_g$. How do the authors set this parameter and what is its impact?
- Would the hard OOD detection task be better qualified as Open-Set Recognition?
- Why the comparison with ZOC is performed using ImageNet-200 and not ImageNet-1k as in Table 2? A Homogeneous comparison setup would greatly enhance the paper quality.

**References**

Lee, K., Lee, K., Lee, H., & Shin, J. (2018). A Simple Unified Framework for Detecting Out-of-Distribution Samples and Adversarial Attacks. Advances in Neural Information Processing Systems, 31.

Sehwag, V., Chiang, M., & Mittal, P. (2022, February 10). SSD: A Unified Framework for Self-Supervised Outlier Detection. International Conference on Learning Representations.

---

> ### Author Response · Authors · 2023-11-19
>
> >**W1:** Assumption about CLIP's latent space The assumption that ID image and corresponding ID label embeddings are close in CLIP's latent space is actually false. The contrastive learning strategy of CLIP is based on cosine similarity and thus textual and visual embeddings are actually on different manifolds see (Weixin Liang et al. 2022) for more details. Using ID labels as proxies for the ID images may thus produce unexpected results. In my understanding, the negative labels are compared to the images using cosine similarity, and thus having them close to ID textual embeddings still makes sense but the authors should be cautious here.
>
> **A1:** Thank you for this valuable comment. We apologize for the inappropriate description. As mentioned in the question, images and text belong to different manifolds in the embedding space of CLIP, so they cannot be considered *close*. We will revise this description in the last paragraph of Section 3.1.
>
> Besides, we would reiterate our OOD detection setting based on CLIP-like VLM models. We only work with a pre-trained CLIP-like model and a specific set of class labels or names for a zero-shot classification task. And the ID images are not available under this zero-shot setting. Hence，using ID labels as the proxies of ID images for negative label selection is intuitive, and it is experimentally verified to be effective in our method. We believe that not relying on the accessibility of ID images allows our method to be used in a wider range of scenarios and is more consistent with the working conditions of CLIP-like models.

---

> > ### Author Response · Authors · 2023-11-19
> >
> > >**W2:** Comparison with related work The method is close in spirit to CLIPN (Wang et al. 2023) as well as ZOC (Esmaeilpour et al 2022). The former fine-tunes CLIP to incorporate negative prompts to assess the probability of a concept not being present in the image. An extensive comparison with CLIPN would be overkill but some discussion about it would be welcome. The second also performs zero-shot OOD using maximum softmax probability on the labels extended with the object detected in the images. Comparisons with ZOC are presented in the supplementary but would better belong in the main paper IMO as it is one of the few recent zero-shot OOD detection baselines. To be fair this comparison should be provided with the same set-up as in Table 2.
> > > **Q4:** Why the comparison with ZOC is performed using ImageNet-200 and not ImageNet-1k as in Table 2? A Homogeneous comparison setup would greatly enhance the paper quality.
> >
> > **A2:** Thanks for your suggestions! We will add some discussions with CLIPN and ZOC in the updated version.
> >
> > **Discussion about CLIPN:**
> >
> > CLIPN utilizes additional datasets to train a text encoder that can understand negative prompts. It discriminates OOD data by comparing the similarity differences between the outputs of two text encoders and the image encoder. **On the other hand, our method does not require additional data or training.** It only introduces negative labels from the corpus to accomplish zero-shot OOD detection tasks. Therefore, in terms of the idea, both methods aim to introduce negative semantics of the ID category to reject inputs unrelated to the ID category. In terms of computational complexity, our method is slightly higher than CLIPN, but it is almost negligible compared to the computational complexity of CLIP itself. In terms of specific performance, our method has a higher average performance on the ImageNet-1k benchmark. Moreover, compared to CLIPN and NegLabel on CLIP-B/16 and CLIP-B/32, our method NegLabel is more robust to model architectures.
> >
> > |        Method        | iNaturalist |       |  SUN  |       | Places |       | Textures |       | Average |       |
> > |:--------------------:|:-----------:|:-----:|:-----:|:-----:|:------:|:-----:|:--------:|:-----:|:-------:|:-----:|
> > |                      |    AUROC    | FPR95 | AUROC | FPR95 |  AUROC | FPR95 |   AUROC  | FPR95 |  AUROC  | FPR95 |
> > |   CLIPN (CLIP-B/32)  |    94.67    | 28.75 | 92.85 | 31.87 |  86.93 | 50.17 |   87.68  | 49.49 |  90.53  | 40.07 |
> > |   CLIPN (CLIP-B/16)  |    95.27    | 23.94 | 93.93 | 26.17 |  90.93 | 40.83 |   92.28  | 33.45 |  93.10  | 31.10 |
> > | NegLabel (CLIP-B/32) |    99.11    |  3.73 | 95.27 | 22.48 |  91.72 | 34.94 |   88.57  | 50.51 |  93.67  | 27.92 |
> > | NegLabel (CLIP-B/16) |    99.49    |  1.91 | 95.49 | 20.53 |  91.64 | 35.59 |   90.22  | 43.56 |  94.21  | 25.40 |
> >
> > **Discussion about ZOC:**
> >
> > As for ZOC, the authors do not report the performances on ImageNet-1k benchmark, and there is no third-party implementation. We think it is mainly because the COCO-pretrained image captioner can not handle the large-scale benchmark ImageNet-1k with 1,000 categories. We try to reproduce an upgraded version of ZOC on ImageNet-1k benchmark. We leveraged the capabilities of the multimodal model LLaVA [1] and ChatGPT to migrate ZOC to large-scale data sets. The idea comes from MCM's discussion and reproduction of ZOC.
> >
> > We first used LLaVA to describe each sample and extract the entities contained in the description. Then we encountered a difficulty. LLaVA's descriptions of a large number of ID samples were slightly different from their corresponding ImageNet-1k labels, such as dog v.s. Alaskan Malamute. As a result, the ID label filtering policy cannot take effect. In order to overcome this problem, we collected all entities generated by LLaVA and used ChatGPT to filter words similar to the ImageNet-1k label. The remaining words are treated as candidate OOD labels of ZOC. The performances of the upgraded version of ZOC are shown in the table below.
> >
> > |  Method  | iNaturalist |          |    SUN    |           |   Places  |           |  Textures |           |  Average  |           |
> > |:--------:|:-----------:|:--------:|:---------:|:---------:|:---------:|:---------:|:---------:|:---------:|:---------:|:---------:|
> > |          |    AUROC    |   FPR95  |   AUROC   |   FPR95   |   AUROC   |   FPR95   |   AUROC   |   FPR95   |   AUROC   |   FPR95   |
> > |    ZOC   |    86.09    |   87.30  |   81.20   |   81.51   |   83.39   |   73.06   |   76.46   |   98.90   |   81.79   |   85.19   |
> > | NegLabel |  **99.49**  | **1.91** | **95.49** | **20.53** | **91.64** | **35.59** | **90.22** | **43.56** | **94.21** | **25.40** |
> >
> > In addition, since the experimental settings of CLIPN and ZOC are different from NegLabel, we will update the comparison of their experimental results in Table 1 and some discussions in the Appendix A.

---

> > > ### Author Response · Authors · 2023-11-19
> > >
> > > >**W3:** Misleading baseline presentation Even if the MSP, ODIN, or Energy Logit baseline are generally used on models fine-tuned on a downstream task, one could also use these detectors in a zero-shot setting. Classifying them in purely zero-shot can be misleading eg. MCM can be seen as zero-shot MSP on CLIP's output probabilities.
> > >
> > > **A3:** Thanks for your suggestions! We will add some zero-shot baselines in Table 1.
> > >
> > > >**W4:** Short Related Work The related work section is relatively short, especially for such a prolific field as OOD detection. I don't think it is really detrimental to the paper but a discussion on the sense of OOD detection for foundation models trained on vast and various data would be welcome.
> > >
> > > **A4:** Thanks for your suggestions! We will add some related works in Section 5 in the updated paper (before the author-reviewer deadline).
> > >
> > > > **Q1:** I wonder why the authors did not add the simple and strong baseline Mahaloanobis (Lee et al. 2018) /SSD (Sehwag et al. 2022).
> > >
> > > **A5:** Thanks for your suggestions! We will add Mahaloanobis as the baseline in Table 1.
> > >
> > > SSD is based on self-supervised learning, and our baselines are based on the pretrained or finetuned CLIP. Therefore, they can not be compared in a fair way.
> > >
> > > > **Q2:** No ablations are provided on n_g. How do the authors set this parameter and what is its impact?
> > >
> > > **A6:** We apologize for this misunderstanding, and we have provided the ablation experiments on $n_g$ in Table 16 of the appendix.
> > >
> > >
> > > > **Q3:** Would the hard OOD detection task be better qualified as Open-Set Recognition?
> > >
> > > **A7:** This is a good question! The hard OOD detection task is set by MCM and we follow it. We think using different subsets of ImageNet as ID and OOD datasets (e.g., ImageNet10 v.s. ImageNet 100) is similar to the setting of OSR. But the setting of the spurious OOD detection task, i.e., WaterBirds v.s. several Spurious OOD, is quite different from OSR.
> > >
> > > **Reference**
> > >
> > > [1] Liu, Haotian, et al. "Visual Instruction Tuning." NeurIPS, 2023.

---

> > > > ### Comment · Reviewer_RPtB · 2023-11-22
> > > >
> > > > I would like to thank the authors for their extended answers. Most of my concerns have been addressed.
> > > >
> > > > Concerning the first point, I agree that using only OOD concepts is a strength of the method. However, I do not see it as a novelty as it was already performed in MCM or Fort et al, 2021. I am still not completely satisfied with the formulation as in my sense the prompts are not used as proxies of the ID images in the paper (and that is a good thing). It makes sense to search for OOD concepts based on the ID ones **but** it does not make sense to see these OOD concepts as pseudo-OOD images.
> > > >
> > > > Moreover, upon rereading, Fort et al, 2021, I see that the last section is the same as the proposed NegLabel score. It seems that the novelty of NegLabel rather relies on the mining algorithm and the grouping strategy than the OOD scorer itself which is not explicit in the contributions. With this in mind, comparisons with this baseline would have been appreciated.

---

> > > > > ### Author Response · Authors · 2023-11-22
> > > > >
> > > > > >Q1: Concerning the first point, I agree that using only OOD concepts is a strength of the method. However, I do not see it as a novelty as it was already performed in MCM or Fort et al, 2021. I am still not completely satisfied with the formulation as in my sense the prompts are not used as proxies of the ID images in the paper (and that is a good thing). It makes sense to search for OOD concepts based on the ID ones but it does not make sense to see these OOD concepts as pseudo-OOD images.
> > > > >
> > > > > **A1:** Thanks for pointing out this issue. We acknowledge that it is not proper to say "using ID labels as the proxies of ID images". Because as the reviewer said in the comment, using ID labels as proxies for ID images means that the selected Negative Labels are proxies for pseudo-OOD images, which lacks theoretical support and is inaccurate. We would like to thank the Reviewer RPtB for the important comments that helped us improve the rigor and conciseness of the paper.
> > > > >
> > > > > As a result, we will state in the paper (especially the last paragraph of Section 3.1) that we use ID labels to search for OOD concepts as Negative Labels that are far away from ID labels.
> > > > >
> > > > > Besides, we would claim that we do not regard zero-shot OOD detection as our contribution, and we just follow the settings of MCM. We believe that retaining the zero-shot characteristics of CLIP-like VLMs is more in line with their deployment scenarios.
> > > > >
> > > > > >Q2: Moreover, upon rereading, Fort et al, 2021, I see that the last section is the same as the proposed NegLabel score. It seems that the novelty of NegLabel rather relies on the mining algorithm and the grouping strategy than the OOD scorer itself which is not explicit in the contributions. With this in mind, comparisons with this baseline would have been appreciated.
> > > > >
> > > > > **A2:** Thanks for the comment. Fort et al, 2021 proposed score=p(in|x) in the probability space. In the Section 3.2, we propose a general form of OOD score with negative labels, which is not limited to probability space and softmax score. In Section A.5.1, we provide several implementations of the score. We will cite Fort et al, 2021 in Section 3.2 and explain its relation to Equation 6.
> > > > >
> > > > > Fort et al, 2021 use the ground-truth OOD labels for OOD detection, which is inconsistent with the OOD detection setting in real scenarios. It may not be appropriate and fair to compare Fort et al, 2021 in Table 1. As an alternative, we will provide the performance of Fort et al, 2021 on ImageNet-1k benchmark in Appendix A.7 and discuss it.
> > > > >
> > > > > In the original implementation of Fort et al, 2021, they directly apply softmax function on the cosine similarities between image and text embeddings, i.e., take temperature $\tau=1$ in Equation 6 in our paper. However, we observe that its performance on the ImageNet-1k benchmark is not as expected because the difference between positive matching and negative matching is too small, as mentioned in the discussion about the temperature coefficient of NegLabel Score in Section 3.2 of our paper. We replace the temperature by $\tau=0.01$ and achieved satisfactory results, with AUROC of 96.88\% and FPR95 of 14.69\%, exceeding all existing results (including zero-shot and non-zero-shot OOD detectors) reported on the ImageNet-1k benchmark. But we remind again that now the OOD detector works in an oracle situation because all OOD labels are added to the negative labels as known information.
> > > > >
> > > > > |      Method      |        | iNaturalist |       |   SUN  |        | Places |        | Textures |        | Average |        |
> > > > > |:----------------:|:------:|:-----------:|:-----:|:------:|:------:|:------:|:------:|:--------:|:------:|:-------:|:------:|
> > > > > |                  | $\tau$ |    AUROC    | FPR95 |  AUROC |  FPR95 |  AUROC |  FPR95 |   AUROC  |  FPR95 |  AUROC  |  FPR95 |
> > > > > |     NegLabel     |  0.01  |    99.49    | 1.91  | 95.49  | 20.53  | 91.64  | 35.59  |  90.22   | 43.56  |  94.21  | 25.40  |
> > > > > | Fort et al, 2021 |    1   |    98.48    | 7.73  | 85.41  | 75.54  | 76.42  | 86.14  |  79.33   | 87.39  |  84.91  | 64.20  |
> > > > > | Fort et al, 2021 |  0.01  |    99.57    | 1.47  | 98.37  |  7.11  | 94.28  | 25.32  |  95.30   | 24.88  |  96.88  | 14.69  |
> > > > >
> > > > > Thanks again to the reviewer for the responses and we are willing to discuss if there are further questions.

---

> > > > > > ### Comment · Reviewer_RPtB · 2023-11-23
> > > > > >
> > > > > > I am satisfied with the authors' rebuttal and I am willing to increase my rating.

---

### Author Response · Authors · 2023-11-21
**Summary of updated content and revisions in the paper**

We would like to express our heartfelt gratitude to the reviewers for dedicating their time and effort to review our paper. In response to their valuable feedback, we have implemented various revisions to our work, which are clearly marked in blue within the revised paper. The key updates are summarized as follows:

* We revise this description in the last paragraph of Section 3.1. Thanks for the suggestions of Reviewer RPtB.
* We update the comparison with ZOC and CLIPN in Table 1 and some discussions in the Appendix B. Thanks for the suggestions of Reviewer RPtB.
* We add some zero-shot baselines in Table 1. Thanks for the suggestions of Reviewer RPtB.
* We revise the comments in Algorithm 1 and revise other algorithms to keep the naming strategy consistent. Thanks for the suggestions of Reviewer JPmP and 9gus.
* More analysis about the high performance on iNaturalist is added in Section 4 and Appendix B.1. Thanks for the suggestions of Reviewer 9gus.
* We provide more details in the main text to help readers better understand the general case in B.3. Thanks for the suggestions of Reviewer JPmP, 9gus and dUt1.
* We reorganize the content in Section 4. Thanks for the suggestions of Reviewer 9gus and dUt1.
* More similarity choices in Eq. (5)  are discussed in the Appendix A.5.6. Thanks for the suggestions of Reviewer dUt1.

---

### Meta-Review · Area_Chair_W6qd · 2023-12-04

**Metareview:**

All reviewers appreciate the contribution of the paper; the rebuttal addressed the concerns well.

**Justification For Why Not Higher Score:**

no scores higher than 8

**Justification For Why Not Lower Score:**

no scores lower than 6

---

### Decision · Program_Chairs · 2024-01-16

Accept (spotlight)